# Doubly Robust Fusion of Many Treatments for Policy Learning

Ke Zhu [1 2 *]   Jianing Chu [3 *]   Ilya Lipkovich [4]   Wenyu Ye [4]   Shu Yang [1]

## Abstract

Individualized treatment rules/recommendations (ITRs) aim to improve patient outcomes by tailoring treatments to the characteristics of each individual. However, when there are many treatment groups, existing methods face significant challenges due to data sparsity within treatment groups and highly unbalanced covariate distributions across groups. To address these challenges, we propose a novel calibration-weighted treatment fusion procedure that robustly balances covariates across treatment groups and fuses similar treatments using a penalized working model. The fusion procedure ensures the recovery of latent treatment group structures when either the calibration model or the outcome model is correctly specified. In the fused treatment space, practitioners can seamlessly apply state-of-the-art ITR learning methods with the flexibility to utilize a subset of covariates, thereby achieving robustness while addressing practical concerns such as fairness. We establish theoretical guarantees, including consistency, the oracle property of treatment fusion, and regret bounds when integrated with multi-armed ITR learning methods such as policy trees. Simulation studies show superior group recovery and policy value compared to existing approaches. We illustrate the practical utility of our method using a nationwide electronic health record-derived de-identified database containing data from patients with Chronic Lymphocytic Leukemia and Small Lymphocytic Lymphoma.

## 1. Introduction

In precision medicine, individualized treatment rules (ITRs) aim to optimize patient outcomes by tailoring treatment recommendations to individuals based on their characteristics. This personalization is essential because treatment effects can vary across individuals. Developing a rule that recommends the most effective treatment for each person requires accounting for these variations in a systematic and data-driven manner. For settings with two or multiple treatments, numerous machine learning approaches have been developed for estimating ITRs, commonly referred to as ITR learning or policy learning. These methods can be broadly classified into two categories. The first involves modeling treatment outcomes, as in Q-learning (Watkins & Dayan, 1992; Qian & Murphy, 2011; Song et al., 2015), A-learning (Murphy, 2003; Shi et al., 2018), or D-learning (Qi & Liu, 2018), where the focus is on estimating the expected outcome under each treatment to derive the optimal rule. The second class of methods directly optimizes the value function, which measures the expected outcome under a given decision rule (Zhang et al., 2012; Zhao et al., 2012; Athey & Wager, 2021). In these methods, inverse propensity score weighting (IPW) or augmented IPW (AIPW) estimators are employed to evaluate the value function, and the optimal rule is identified by maximizing the value function over a class of decision functions, such as linear ITRs (Zhang et al., 2012; Zhao et al., 2012) or tree-based ITRs (Zhang et al., 2015; Laber & Zhao, 2015; Athey & Wager, 2021).

However, these approaches encounter significant challenges when the number of treatment levels becomes large (Rashid et al., 2021). When treatment levels are numerous, data is often sparse within each treatment group, making it difficult to estimate treatment effects accurately. Additionally, covariate shifts across treatment groups exacerbate instability, particularly when balancing methods like IPW are used. This instability is further amplified in underrepresented treatment groups, where propensity scores are small and highly variable.

A key insight to address these challenges lies in recognizing that many treatments share commonalities (Ma et al., 2022; 2023). For example, different pharmaceutical companies may develop treatments targeting the same disease mechanisms or symptoms, resulting in similar effects across

---

*Equal contribution  [1]Department of Statistics, North Carolina State University, Raleigh, NC 27695, U.S.A. [2]Department of Biostatistics and Bioinformatics, Duke University, Durham, NC 27710, U.S.A. [3]Amazon (This work was done prior to joining Amazon) [4]Eli Lilly & Company, Indianapolis, IN 46285, U.S.A.. Correspondence to: Shu Yang <syang24@ncsu.edu>.

*Proceedings of the 42nd International Conference on Machine Learning*, Vancouver, Canada. PMLR 267, 2025. Copyright 2025 by the author(s).

treatments. By grouping such treatments into clusters, we can reduce the effective dimensionality of the treatment space. Treatments within the same group can be treated as equivalent, enabling the application of efficient multi-armed ITR learning methods to the grouped treatments.

Despite its potential, the task of treatment fusion introduces its own challenges due to sparse data and unbalanced covariate distributions. Sparse data within treatment groups necessitates the use of simple linear working models to prevent overfitting. However, these models risk misspecification, which can lead to biased fusion results. Additionally, severe covariate shifts across groups make traditional balancing methods like IPW unreliable, especially for treatments with very small sample sizes.

To overcome these difficulties, we propose a novel procedure called calibration-weighted treatment fusion. This method uses calibration weighting (Lee et al., 2023; Wu & Yang, 2023) to robustly balance covariates across treatment groups, addressing the instability of traditional balancing techniques. The calibrated weights are then used in a penalized working model with fused Lasso (Tibshirani et al., 2005) to group treatments based on their effects. Our procedure is doubly robust, meaning that the true latent group structure can be recovered as long as either the calibration weighting model or the outcome model is correctly specified. This robustness significantly enhances both the efficiency and reliability of treatment fusion compared to existing methods.

After performing data-driven fusion, practitioners can transparently review the grouping results and seamlessly apply state-of-the-art multi-armed ITR learning methods on the grouped treatments, such as policy trees (Zhou et al., 2023), to align with their application contexts. This approach not only robustly reduces the dimensionality of the treatment space but also ensures flexibility, interpretability, and improved policy learning outcomes. By addressing the challenges of many treatments, our method provides a robust and practical framework for advancing precision medicine.

### 1.1. Related work

In this paper, we study settings with a large action space induced by a single discrete treatment variable with many levels (Saito & Joachims, 2022; Saito et al., 2023; Peng et al., 2023; Sachdeva et al., 2024; Aouali et al., 2024). Related work considers policy evaluation or learning under alternative treatment structures.

**Combination treatments.** Some studies consider treatments formed by combinations of multiple variables (Liang et al., 2018; Agarwal et al., 2023; Xu et al., 2024a;b). Agarwal et al. (2023) propose synthetic combinations that impute counterfactuals via low-rank matrix completion under struc-

tural assumptions. Gao et al. (2024) used a low-rank tensor with block structure to fuse treatments. In contrast, we assume a group structure among treatment levels and apply calibration-weighted fused lasso. These approaches are complementary, targeting different structural assumptions.

**Continuous treatments.** Other methods focus on continuous treatments (e.g., Chernozhukov et al., 2019; Cai et al., 2021). Cai et al. (2021) also explores action grouping in a continuous setting. These approaches differ methodologically from ours, which focuses on discrete actions.

**Complex treatments.** Recent work studies complex treatment types such as images, text, or chemical structures (Kaddour et al., 2021; Nilforoshan et al., 2023; Schweisthal et al., 2023; Marmarelis et al., 2024). Schweisthal et al. (2023) address large treatment spaces induced by high-dimensional continuous variables using neural networks and constrained optimization, focusing on regions with sufficient overlap. In contrast, our method targets the entire population, is easy to implement, compatible with existing algorithms, and interpretable.

## 2. Preliminaries

We consider a $K$-armed setting where the treatment

$$A \in \mathcal{A} := \{1, 2, \ldots, K\}.$$

Let $X \in \mathcal{X} \subseteq \mathbb{R}^p$ denote a vector of covariates, and $Y \in \mathbb{R}$ denote the observed outcome of interest. We assume that larger values of $Y$ are preferred by convention. The observed data $(Y_i, A_i, X_i)$ are assumed to be independent and identically distributed. The potential outcomes $Y(a)$, $a \in \mathcal{A}$, represent the outcomes that would be observed if a subject received treatment $a$. The following standard assumptions in causal inference are made (Rubin, 1978).

**Assumption 2.1** (Identification). (i) Consistency: $Y = Y(A)$. (ii) Unconfoundedness: $Y(a) \perp\!\!\!\perp A \mid X, \forall a \in \mathcal{A}$. (iii) Positivity: $0 < \mathbb{P}(A = a \mid X = x) < 1$ for all $a \in \mathcal{A}$.

An individualized treatment rule (ITR) is a decision function $d(\cdot) : \mathcal{X} \to \mathcal{A}$, which maps the covariate space to the treatment space. For any arbitrary ITR $d(\cdot)$, the corresponding potential outcome is defined as $Y(d(X))$, which would be observed if a randomly chosen individual were assigned treatment according to $d(\cdot)$, i.e., $A = d(X)$. The value function under $d(\cdot)$ is then defined as the expectation of $Y(d(X))$, i.e., $V(d) := \mathbb{E}\{Y(d(X))\}$. Let the propensity score be

$$\pi_a(x) = \mathbb{P}(A = a \mid X = x),$$

and the outcome mean function be

$$\mu_a(x) = \mathbb{E}\{Y(a) \mid X = x\}.$$

Under Assumption 2.1, the value function $V(d)$ can be identified using observed data through inverse propensity score

weighting (IPW): $V(d) = \mathbb{E}\left[Y\mathbb{I}\{A = d(X)\}\pi_A^{-1}(X)\right]$. Suppose $\mathcal{D}$ is a class of ITRs of interest, such as linear ITRs (Zhang et al., 2012; Zhao et al., 2012; Cheng & Yang, 2024) or tree-based ITRs (Zhang et al., 2015; Laber & Zhao, 2015; Athey & Wager, 2021). The optimal ITR is defined as $d^*(X) := \operatorname{argmax}_{d \in \mathcal{D}} V(d)$. The complexity of $\mathcal{D}$ increases exponentially with $K$. Consequently, existing literature often assumes that $K$ is fixed (Zhou et al., 2023). However, in practice, the treatment dimension $K$ may be high (Rashid et al., 2021), making the task of learning $d^*(X)$ significantly more challenging.

To address this challenge, a key insight is that certain treatments, such as those in drug development targeting similar disease symptoms and mechanisms, may yield comparable or identical outcomes (Ma et al., 2022; 2023). This observation suggests a group structure.

**Definition 2.2** (Oracle group structure). $\mathcal{A} = \cup_{b=1}^M \mathcal{G}_b^*$, where $\mathcal{G}_b^*$'s are disjoint sets satisfying: (i) $\mu_a(X) = \mu_{a'}(X)$ for $a, a' \in \mathcal{G}_b^*$ and (ii) $\mu_a(X) \neq \mu_{a'}(X)$ for $a \in \mathcal{G}_b^*$, $a' \in \mathcal{G}_{b'}^*$ with $b \neq b'$.

*Remark* 2.3. We define the oracle group structure by exact equality of $\mu_a(X)$ to ensure identifiability and interpretability. While seemingly restrictive, this serves as a natural basis for grouping similar treatments and enables formal guarantees. In practice, the fused lasso penalization we use allows for grouping treatments with approximately equal effects by tolerating small differences due to sampling variability. The explicit gap tolerance required for recovery is provided in Remark 3.11. Overall, when no exact group structure exists, fusion trades variance for bias, and this trade-off can improve performance when data are limited.

Motivated by this latent structure, we can first learn a group mapping $\delta : \mathcal{A} \to \mathcal{B} := \{1, 2, \ldots, M\}$, and subsequently learn the grouped ITR $d^{\mathcal{B}}(\cdot) : \mathcal{X} \to \mathcal{B}$ by using established multi-armed policy learning methods. Since $M$ is smaller than $K$ after grouping, learning the grouped optimal ITR $d^{\mathcal{B}*} := \operatorname{argmax}_{d^{\mathcal{B}} \in \mathcal{D}^{\mathcal{B}}} V(d)$ becomes more efficient, where $\mathcal{D}^{\mathcal{B}}$ is a class of ITRs $d^{\mathcal{B}}$. After obtaining $\hat{d}_{\mathcal{B}}$, for any $X$ such that $\hat{d}_{\mathcal{B}}(X) = b$, we define $\hat{d}(X)$ as randomly selecting one $a$ such that $a \in \mathcal{G}_b^*$.

Therefore, the primary objective of this paper is to recover the true group structure $\cup_{b=1}^M \mathcal{G}_b^*$ through data-driven fusion. This task is particularly challenging due to the sparsity of data within treatment groups caused by a large $K$, which hinders the accurate estimation of $\mu_a(X)$ using complex models beyond linear ones. Additionally, some treatment groups have very low propensity scores, leading to high variability in conventional inverse propensity score balancing methods. Both model misspecification and covariate shifts can introduce bias, resulting in poor treatment fusion and suboptimal policy learning. To address these challenges, we propose a calibration weighting method that robustly balances covariates across treatments. Since $\mu_a(x) = \mu_{a'}(x)$ implies that the linear projections of $Y(a)$ and $Y(a')$ are equal, we employ a penalized linear working model to perform treatment fusion, achieving double robustness when combined with calibration weighting. After treatment grouping, any state-of-the-art ITR learning method, such as policy trees, can be adopted, allowing for flexible outcome modeling and superior policy learning.

# 3. Methodology

## 3.1. Calibration-Weighted Treatment Fusion

Since the number of groups $M$ and the group structure $\cup_{b=1}^M \mathcal{G}_b^*$ are both unknown, we employ fused Lasso to jointly determine $M$ and the partition. Specifically, we consider the following working linear model:

$$Y = M_0(X) + \sum_{a \in \mathcal{A}} \mathbb{I}(A = a)X^\top \zeta_a + \epsilon,$$

$$\text{s.t.} \quad \sum_{a \in \mathcal{A}} \mathbb{I}(A = a)X^\top \zeta_a = 0,$$

where the redundant function $M_0(X)$ represents the main effect of treatments, and $X^\top \zeta_a$ captures the interaction effect between treatment $a$ and the covariates. The sum-to-zero constraint on the interaction terms ensures the identifiability of the regression function. The main effect function $M_0(X)$ can be estimated using weighted parametric or nonparametric regression models. As this is not the focus of our paper, we assume it has been accurately estimated and define the transformed outcome as $\tilde{Y} = Y - M_0(X)$.

To estimate and group $\zeta_a$'s, we consider the following optimization problem, which imposes a pairwise fusion penalty on each pair of treatment-specific parameters:

$$\min_{\zeta} \left\{ \frac{1}{n} \sum_{a \in \mathcal{A}} \sum_{i:A_i=a} \mathcal{L}\left(\tilde{Y}_i, X_i^\top \zeta_a\right) + \sum_{1 \leqslant a < a' \leqslant K} p_{\lambda_n}\left(\|\zeta_a - \zeta_{a'}\|_1\right) \right\}, \quad (1)$$

where $\mathcal{L}(\cdot, \cdot)$ is a prespecified loss function that measures the goodness of fit, $\|\cdot\|_1$ denotes the $\ell_1$ norm of a vector, $p_{\lambda_n}$ is a penalty function that encourages the fusion of $\hat{\zeta}_a$'s into groups, and $\lambda_n$ is the tuning parameter, which can be selected by multiple model selection criteria, such as Bayesian information criterion (BIC) (Schwarz, 1978) or extended BIC (EBIC) (Chen & Chen, 2008).

However, the applicability of the objective function (1) is limited to scenarios where either (i) the true outcome function $\mu_a(x)$ is linear or (ii) the covariate distributions are identical across all treatment groups. In cases where these conditions do not hold, the estimated $\zeta_a$'s obtained from

(1) may deviate from the true group structure $\cup_{b=1}^{M} \mathcal{G}_b^*$. In observational studies, variations in covariate distributions across different treatment groups often prevent these groups from accurately representing the entire population. To mitigate this issue, a preliminary step involves reweighting the samples within each treatment group so that the weighted samples better reflect the overall population. To achieve this, we propose the following calibration weighting approach.

For each treatment group $a$, we aim to assign weights $\{w_i : A_i = a\}$ to calibrate the covariate distribution of the group to match the overall sample mean $\bar{X}$. This is achieved by solving the following optimization problem for each $a \in \mathcal{A}$:

$$\min_{w_i, i:A_i=a} \sum_{i:A_i=a} h_\gamma(w_i),$$

$$\text{s.t.} \quad \sum_{i:A_i=a} w_i X_i = \bar{X}, \quad \sum_{i:A_i=a} w_i = 1, \qquad (2)$$

where $h_\gamma(w)$ quantifies the discrepancy between the calibration weights and the uniform distribution $n_a^{-1}$, with $n_a$ denoting the sample size of treatment group $a$. The function $h_\gamma(w)$ can be chosen from the Cressie and Read family of discrepancies (Cressie & Read, 1984), defined as:

$$\sum_{i:A_i=a} h_\gamma(w_i) = \sum_{i:A_i=a} \{\gamma(\gamma+1)\}^{-1}\{(n_a w_i)^{\gamma+1} - 1\}.$$

For example, minimizing $\sum_{i:A_i=a} h_{-1}(w_i)$ is equivalent to maximizing $\sum_{i:A_i=a} \log(w_i)$, leading to the maximum empirical log-likelihood objective function. Minimizing $\sum_{i:A_i=a} h_0(w_i)$ is equivalent to maximizing $-\sum_{i:A_i=a} w_i \log(w_i)$, leading to the maximum empirical exponential likelihood or entropy.

Let $\widehat{w}_i$ denote the calibrated weights solved by (2) for individual $i$. Using these weights, the calibrated objective function becomes:

$$\min_\zeta \left\{ \frac{1}{n} \sum_{a \in \mathcal{A}} \sum_{i:A_i=a} \widehat{w}_i \mathcal{L}\left(\tilde{Y}_i, X_i^\top \zeta_a\right) \right.$$
$$\left. + \sum_{1 \leqslant a < a' \leqslant K} p_{\lambda_n}\left(\|\zeta_a - \zeta_{a'}\|_1\right) \right\}. \qquad (3)$$

The entire procedure is summarized in Algorithm 1, considering the least squares loss function as an example.

### 3.2. Double Robustness of Treatment Fusion

**Theory roadmap.** This section provides theoretical guarantees for Algorithm 1. In Section 3.2.1, we represent the oracle group structure $\cup_{b=1}^{M} \mathcal{G}_b^*$ via the linear projection of potential outcomes onto the covariate space, without requiring the linear model to be correctly specified. Under the completeness Assumption 3.1, recovering the oracle group

---

**Algorithm 1:** Calibration-Weighted Treatment Fusion

**Input:** Data $\{(X_i, A_i, Y_i)\}_{i=1}^n$.

**for** $a = 1, \ldots, K$ **do**
  Solve calibration weights $\hat{w}_i$ by optimizing:

  $$\min_{w_i, i:A_i=a} \sum_{i:A_i=a} h_\gamma(w_i),$$

  $$\text{s.t.} \quad \sum_{i:A_i=a} w_i X_i = \bar{X}, \quad \sum_{i:A_i=a} w_i = 1,$$

Solve $\widehat{\zeta}$ by weighted fused Lasso:

$$\min_{\zeta=(\zeta_1^\top, \ldots, \zeta_K^\top)^\top \in \mathbb{R}^{Kp}} \{L_n(\zeta) + P_n(\zeta)\},$$

$$L_n(\zeta) = \frac{1}{2n} \sum_{a \in \mathcal{A}} \sum_{i:A_i=a} \widehat{w}_i \left(\tilde{Y}_i - X_i^\top \zeta_a\right)^2,$$

$$P_n(\zeta) = \sum_{1 \leqslant a < a' \leqslant K} p_{\lambda_n}\left(\|\zeta_a - \zeta_{a'}\|_1\right).$$

Forming groups $\delta(a) = \delta(a')$ if $\widehat{\zeta}_a = \widehat{\zeta}_{a'}$.
**Output:** Group mapping $\delta : \mathcal{A} \to \mathcal{B}$.

---

structure is equivalent to recovering the projection vectors. In Section 3.2.2, we establish the convergence of the oracle estimator for the projection vectors under doubly robust and regularity conditions (Theorem 3.8). In Section 3.2.3, we show that the oracle estimator is a local minimizer of the objective function in Algorithm 1 (Theorem 3.12). Taken together, these results imply that Algorithm 1 consistently recovers the oracle group structure. Technical clarifications are provided in remarks and can be skipped by readers less interested in such details.

#### 3.2.1. REPRESENTATION OF ORACLE GROUP STRUCTURE

For $a \in \mathcal{A}$, let $\tilde{Y}(a) := Y(a) - M_0(X)$ denote the transformed potential outcome. We project $\tilde{Y}(a)$ onto the linear space spanned by $X$ and denote the projection vector by

$$\zeta_a^* := \underset{\zeta \in \mathbb{R}^p}{\operatorname{argmin}} \, \mathbb{E}\left\{\tilde{Y}(a) - X^\top \zeta\right\}^2, \qquad (4)$$

where $X$ includes the intercept term. Solving (4) yields:

$$\mathbb{E}\left[X^\top \left\{\tilde{Y}(a) - X^\top \zeta_a^*\right\}\right] = 0. \qquad (5)$$

We define the projection residual by

$$\varepsilon(a) := \tilde{Y}(a) - X^\top \zeta_a^*. \qquad (6)$$

By (5) and (6), we have

$$\tilde{Y}(a) = X^\top \zeta_a^* + \varepsilon(a), \quad \mathbb{E}\left\{X^\top \varepsilon(a)\right\} = 0. \qquad (7)$$

From the above derivation, condition (7) does not assume a linear relationship between $\tilde{Y}(a)$ and $X$; it holds solely due to the projection (4) and the definition of $\varepsilon(a)$. Condition (7) alone is generally insufficient to guarantee the consistency of the estimated projection vector derived from the unweighted working linear model (1). Ma et al. (2022) assumes a linear relationship between $\tilde{Y}(a)$ and $X$, which essentially imposes a stronger condition:

$$\tilde{Y}(a) = X^\top \zeta_a^* + \varepsilon(a), \quad \mathbb{E}\{\varepsilon(a) \mid X\} = 0. \quad (8)$$

In the following, we show that by using calibration-weighted treatment fusion (3), the consistency results hold if either the calibration weighting (2) is correctly specified or the outcome model (8) is correctly specified, but not necessarily both. Consequently, our approach provides a more robust fusion method against model misspecification.

Since $X$ includes the intercept term, and from (7), we have $\mathbb{E}\{\varepsilon(a)\} = 0$. Therefore,

$$a, a' \in \mathcal{G}_b^* \iff \mu_a(X) = \mu_{a'}(X) \implies \zeta_a^* = \zeta_{a'}^*. \quad (9)$$

To ensure the reverse direction of (9) holds, we impose the following assumption.

**Assumption 3.1** (Completeness)**.** For any function $h(\cdot)$, if $\mathbb{E}Xh(X) = 0$, then $h(X) = 0$ almost surely.

Under Assumption 3.1, we have

$$\zeta_a^* = \zeta_{a'}^* \implies \mu_a(X) = \mu_{a'}(X),$$

which ensures that identifying $\zeta_a^*$ recovers the oracle grouping.

We denote the *group-shared projection vector* as $\boldsymbol{\beta}_b^* := \zeta_a^*$, $\forall a \in \mathcal{G}_b^*$. The transformed potential outcome can then be expressed as:

$$\tilde{Y}(a) = X^\top \zeta_a^* + \varepsilon(a) = X^\top \boldsymbol{\beta}_b^* + \varepsilon(a). \quad (10)$$

If the true group structure $\cup_{b=1}^M \mathcal{G}_b^*$ is known, the data within each group can be pooled to estimate the group-shared projection vector as

$$\widehat{\boldsymbol{\beta}}_b = \min_{\boldsymbol{\beta}_m \in \mathbb{R}^p} \frac{1}{2n} \sum_{i=1}^n \sum_{a \in \mathcal{G}_b^*} \mathbb{I}(A_i = a) \widehat{w}_i \left( \tilde{Y}_i - X_i^\top \boldsymbol{\beta}_m \right)^2.$$

Then, the oracle estimator for the projection vector $\zeta^* = (\zeta_1^{*\top}, \dots, \zeta_K^{*\top})^\top$ can be obtained by expanding $\widehat{\boldsymbol{\beta}}_b$, such that $\widehat{\zeta}^{\text{or}} = (\widehat{\zeta}_1^{\text{or}\top}, \dots, \widehat{\zeta}_K^{\text{or}\top})^\top$, where $\widehat{\zeta}_a^{\text{or}} \equiv \widehat{\boldsymbol{\beta}}_b$ for all $a \in \mathcal{G}_b^*$. In practice, since the true group structure is unknown, the estimated projection vector $\widehat{\zeta}$ is obtained using Algorithm 1. Define the objective function as $Q_n(\zeta) = L_n(\zeta) + P_n(\zeta)$.

### 3.2.2. CONSISTENCY OF ORACLE ESTIMATOR

We establish the convergence of $\widehat{\zeta}^{\text{or}}$ to $\zeta^*$ under the assumptions stated below. Let $C_1$, $C_2$, $C_3$, and $C_4$ denote positive constants. We allow $K$, $M$, and $p$ to grow with $n$, omitting their dependence on $n$ for notational simplicity. We write $a_n \gg b_n$ to denote $b_n = o(a_n)$.

**Assumption 3.2** (Convergence of calibration weight)**.** $\forall i = 1, \dots, n$, $\widehat{w}_i = w_i^* + O_\mathbb{P}(1/\sqrt{n})$ and $C_1 \leq w_i^* \leq C_2$.

**Assumption 3.3** (Doubly robust model assumption)**.** One of the following conditions holds: (i) (Correct calibration weighting) $w_i^* = 1/\pi_{A_i}(X_i)$, or (ii) (Correct outcome model) $\mathbb{E}\{\varepsilon(a) \mid X\} = \mathbb{E}\{\tilde{Y}(a) - X^\top \zeta_a^* \mid X\} = 0$ for any $a \in \mathcal{A}$.

*Remark* 3.4. Assumption 3.2 requires the $\sqrt{n}$-convergence of the working weights $\widehat{w}_i$ to bounded limits $w_i^*$, which are not necessarily the true inverse propensity scores. This typically holds under mild conditions for posited parametric models for weighting, such as the entropy balancing method (see Section A.1 for details). The convergence rate requirement for the weights in the fusion stage may be relaxed by using undersmoothed estimators or advanced doubly robust methods (Chambaz et al., 2012; Ertefaie et al., 2023; Bruns-Smith et al., 2025). Assumption 3.3 requires only that either the calibration weighting model or the outcome model is correctly specified, highlighting the double robustness of our results.

**Assumption 3.5** (Regularity condition for $X$)**.** For any $j = 1, \dots, p$, $n^{-1} \sum_{i=1}^n X_{ij}^2 \leq C_3$. For any $b \in \mathcal{B}$, $\Lambda_{\min}\left( \sum_{i:A_i \in \mathcal{G}_b^*} X_i X_i^\top \right) / N_{\min} \geq C_4$, where $\Lambda_{\min}(\cdot)$ denote the smallest eigenvalue of a matrix, and $N_{\min} := \min_{b \in \mathcal{B}} \sum_{i=1}^n \mathbb{I}\{A_i \in \mathcal{G}_b^*\}$ is the smallest sample size across groups.

**Assumption 3.6** (Sub-Gaussian error)**.** For any $a \in \mathcal{A}$, $\boldsymbol{\varepsilon}(a) := (\varepsilon_1(a), \dots, \varepsilon_n(a))^\top$ has sub-Gaussian tails, that is, $\exists \sigma_\varepsilon > 0$, for any $\boldsymbol{b} \in \mathbb{R}^n$ and $t > 0$, $\mathbb{P}(|\boldsymbol{b}^\top \boldsymbol{\varepsilon}(a) - \mathbb{E}\{\boldsymbol{b}^\top \boldsymbol{\varepsilon}(a)\}| > \|\boldsymbol{b}\|_2 t) \leq 2 \exp(-t^2/2\sigma_\varepsilon^2)$.

*Remark* 3.7. Assumptions 3.5 and 3.6 are typical regularity conditions used in high-dimensional statistics (Wainwright, 2019). Notably, Assumption 3.6 only requires that $\boldsymbol{b}^\top \boldsymbol{\varepsilon}(a)$ concentrates around its expectation. If the outcome model is misspecified and there is covariate shift across treatments, $\mathbb{E}\{\boldsymbol{b}^\top \boldsymbol{\varepsilon}(a)\}$ may not equal zero when $b_i = \mathbb{I}(A_i = a)X_{ij}$, leading to bias. However, by using calibration weighting with $b_i = \mathbb{I}(A_i = a)w_i^* X_{ij}$, we can robustly eliminate this bias if either the calibration weighting model or the outcome model is correctly specified, as shown in Lemma A.1.

**Theorem 3.8** (Consistency of $\widehat{\zeta}^{\text{or}}$)**.** *Suppose Assumptions 2.1, 3.2, 3.3, 3.5, and 3.6 hold. If $Mp/n = o(1)$ and $\sqrt{p\,n \log(n)}/N_{\min} = o(1)$, then for some constant $C > 0$, with probability at least $1 - 2Mp/n - \iota_n$ (where $\iota_n \to 0$ as $n \to \infty$), we have $\|\widehat{\zeta}^{\text{or}} - \zeta^*\|_\infty \leq C\sqrt{p\,n \log(n)}/N_{\min}$.*

*Remark* 3.9. To ensure the consistency of $\widehat{\zeta}^{\mathrm{or}}$, it is required that $\sqrt{p\,n\log(n)} \ll N_{\min} \leq n/M$, which implies that the number of groups must satisfy $M = o\left(\sqrt{n/\{p\log(n)\}}\right)$.

### 3.2.3. ORACLE PROPERTY OF $\widehat{\zeta}$

Next, we establish the oracle property of $\widehat{\zeta}$. To encourage the grouping of similar projection vectors and reduce bias introduced by the penalty, we require the penalty function $p_{\lambda_n}(\cdot) := \lambda_n\rho(\cdot)$ to have a sharp derivative near 0. Moreover, the $\ell_\infty$ distances between the projection vectors of two different groups must be sufficiently large to ensure they can be separated. Thus, we impose the following regularity condition, which is commonly used in high-dimensional statistics (e.g., Ma & Huang, 2017).

**Assumption 3.10** (Penalty function)**.** The penalty function $p_{\lambda_n}(\cdot) = \lambda_n\rho(\cdot)$ is symmetric about 0, satisfies $p_{\lambda_n}(0) = 0$, is differentiable near 0 with $\rho'(t)$ continuous except at finitely many $t$ and $\rho'(0+) = 1$, and becomes constant for $t \geq c\lambda_n/2$ for some $c > 0$. Additionally, $\min_{b\neq b'} \|\boldsymbol{\beta}_b^* - \boldsymbol{\beta}_{b'}^*\|_\infty/c > \lambda_n \gg \phi_n + p\,\phi_n/K_{\min} + \sqrt{n\log(n)}/K_{\min}$, where $\phi_n := C\sqrt{p\,n\log(n)}/N_{\min}$, $C$ is the constant in Theorem 3.8, and $K_{\min} := \min_{b\in\mathcal{B}} |\mathcal{G}_b^*|$ is the smallest number of treatments across groups.

*Remark* 3.11. When (i) the covariate dimension $p$ and the number of groups $M$ are fixed, and (ii) $N_{\min} = \eta_1 n/M$ and $K_{\min} = \eta_2 K/M$, where $\eta_1, \eta_2 \in (0,1]$ are fixed constants, the term $\phi_n + p\,\phi_n/K_{\min} + \sqrt{n\log(n)}/K_{\min}$ is of order $O(\sqrt{n\log(n)}/K)$. In this simplified case, it suffices to assume that $\min_{b\neq b'} \|\boldsymbol{\beta}_b^* - \boldsymbol{\beta}_{b'}^*\|_\infty/c > \lambda_n \gg \sqrt{n\log(n)}/K$.

**Theorem 3.12** (Oracle property of $\widehat{\zeta}$)**.** *Suppose the conditions in Theorem 3.8 and Assumption 3.10 are satisfied. If $Kp/n = o(1)$, there exists a local minimizer $\widehat{\zeta}$ of the objective function $Q_n(\boldsymbol{\zeta})$ such that $\mathbb{P}(\widehat{\zeta} = \widehat{\zeta}^{\mathrm{or}}) \to 1$.*

Combining Theorems 3.8 and 3.12, we demonstrate that minimizing $Q_n(\boldsymbol{\zeta})$ facilitates the recovery of $\boldsymbol{\zeta}^*$, which indicates the group structure $\cup_{b=1}^M \mathcal{G}_b^*$ under the completeness Assumption 3.1.

### 3.3. Multi-armed Policy Learning

After fusing treatments, those within the same group can be treated as identical. This enables our doubly robust treatment fusion procedure to seamlessly integrate with any state-of-the-art multi-armed ITR learning method to identify the optimal grouped ITR $d^{\mathcal{B}*} : \mathcal{X} \to \mathcal{B}$, where $\mathcal{B} = \{1, 2, \ldots, M\}$ represents the set of group indices.

As a concrete example, we combine the doubly robust treatment fusion with the Cross-Fitted Augmented IPW Learning (CAIPWL) approach proposed by Zhou et al. (2023), em-

ploying policy trees as the specific policy class, and review its theoretical results. CAIPWL involves three main steps. First, it estimates the nuisance functions

$$\pi_b(x) := \sum_{a\in\mathcal{G}_b^*} \pi_a(x), \quad \mu_b(x) := \mu_a(x), \forall a \in \mathcal{G}_b^*,$$

using $L$-fold cross-fitting. Next, to evaluate the value of a policy $d^{\mathcal{B}}$, an augmented IPW (AIPW) estimator is used:

$$\hat{V}(d^{\mathcal{B}}) = \frac{1}{n}\sum_{i=1}^n \left\{ \mathbb{I}\{B_i = d^{\mathcal{B}}(X_i)\}\frac{Y_i - \hat{\mu}_{B_i}^{-l(i)}(X_i)}{\hat{\pi}_{B_i}^{-l(i)}(X_i)} + \hat{\mu}_{d^{\mathcal{B}}(X_i)}^{-l(i)}(X_i) \right\}. \quad (11)$$

Finally, $\hat{V}(d^{\mathcal{B}})$ is optimized over a specified policy class $\mathcal{D}^{\mathcal{B}}$, such as a decision tree, to obtain the ITR estimator $\hat{d}^{\mathcal{B}}$. The detailed procedure is outlined in Algorithm 2. Note that $\hat{\mu}_{B_i}^{-l(i)}(X_i)$ and $\hat{\pi}_{B_i}^{-l(i)}(X_i)$ denote the nuisance functions estimated using $L - 1$ folds of data, excluding the $l(i)$-th fold containing the $i$-th unit.

---

**Algorithm 2:** Cross-Fitted AIPW Policy Learning

**Input:** Data $\{(X_i, A_i, Y_i)\}_{i=1}^n$; Group mapping $\delta$.
Mapping the treatment into groups $B_i = \delta(A_i)$.
Split the data into $L$ folds.
**for** $l = 1, \ldots, L$ **do**
    **for** $b = 1, \ldots, M$ **do**
        Fit $\hat{\pi}_b^{-l}(x)$ using rest $L - 1$ folds.
        Fit $\hat{\mu}_b^{-l}(x)$ using rest $L - 1$ folds.
Compute the estimated value of a policy $d^{\mathcal{B}}$ by (11).
Solving $\hat{d}^{\mathcal{B}} = \mathrm{argmax}_{d^{\mathcal{B}}\in\mathcal{D}^{\mathcal{B}}} \hat{V}(d^{\mathcal{B}})$.
**Output:** Optimal policy $\hat{d}^{\mathcal{B}}$.

---

Consider $\mathcal{D}^{\mathcal{B}} = \mathcal{D}_{\mathrm{tree}}^{\mathcal{B}}$ in Algorithm 2, where depth-$D$ trees serve as candidate policies $d^{\mathcal{B}}(\cdot) : \mathcal{X} \to \mathcal{B}$. A depth-$D$ tree maps a covariate vector $X = (X^1, \ldots, X^p) \in \mathcal{X}$ into an action $b \in \mathcal{B}$ by traversing $D - 1$ branch layers followed by a final layer of leaf nodes. Each branch node splits on a covariate $X^j$ at a threshold $l$, directing traversal to the left child if $X^j < l$ and to the right child otherwise. The traversal ends at a leaf node, assigned one of $M$ actions in $\mathcal{B}$, partitioning $\mathcal{X}$ into up to $2^D$ disjoint regions, each associated with an action $b$. Figure 1 shows a depth-5 tree learned from a real data application.

Notably, the covariates used for node splits may be a smaller subset of those used for treatment fusion in Algorithm 1 and for estimating nuisance functions $\mu_b(X)$ and $\pi_b(X)$. While treatment fusion and nuisance function estimation employ a richer covariate set to ensure correct model specification, the decision function $d^{\mathcal{B}}$ prioritizes a smaller, interpretable subset of covariates that are actionable and relevant for

*Table 1.* Summary of definitions for the CAIPWL and policy tree.

| Description | Notation | Definition |
|---|---|---|
| Hamming Distance on $\{x_i\}_{i=1}^n \subset \mathcal{X}$ | $H(d_1^{\mathcal{B}}, d_2^{\mathcal{B}}; \{x_i\}_{i=1}^n)$ | $n^{-1} \sum_{i=1}^n \mathbb{I}\{d_1^{\mathcal{B}}(x_i) \neq d_2^{\mathcal{B}}(x_i)\}$. |
| $\epsilon$-Hamming Covering Number on $\{x_i\}_{i=1}^n \subset \mathcal{X}$ | $N_H(\epsilon, \mathcal{D}^{\mathcal{B}}, \{x_i\}_{i=1}^n)$ | The smallest number $L$ of policies $d_1^{\mathcal{B}}, \ldots, d_L^{\mathcal{B}}$ in $\mathcal{D}^{\mathcal{B}}$, such that $\forall d^{\mathcal{B}} \in \mathcal{D}^{\mathcal{B}}, \exists d_i^{\mathcal{B}} \in \mathcal{D}^{\mathcal{B}}, H(d^{\mathcal{B}}, d_i^{\mathcal{B}}; \{x_i\}_{i=1}^n) \leq \epsilon$. |
| $\epsilon$-Hamming Covering Number | $N_H(\epsilon, \mathcal{D}^{\mathcal{B}})$ | $\sup\{N_H(\epsilon, \mathcal{D}^{\mathcal{B}}, \{x_i\}_{i=1}^m) : m \geq 1, \{x_i\}_{i=1}^m \subset \mathcal{X}\}$. |
| Entropy Integral | $\kappa(\mathcal{D}^{\mathcal{B}})$ | $\int_0^1 \sqrt{\log N_H(\epsilon^2, \mathcal{D}^{\mathcal{B}})} d\epsilon$. |
| Policy value of $d^{\mathcal{B}}$ | $\phi(d^{\mathcal{B}})$ | $\mathbb{I}\{B = d^{\mathcal{B}}(X)\} \frac{Y - \mu_B(X)}{\pi_B(X)} + \mu_{d^{\mathcal{B}}(X)}(X)$. |
| Worst-case variance in evaluating the difference between two policies in $\mathcal{D}^{\mathcal{B}}$ | $V_*$ | $\sup_{d_1^{\mathcal{B}}, d_2^{\mathcal{B}} \in \mathcal{D}^{\mathcal{B}}} \mathbb{E}\{\phi(d_1^{\mathcal{B}}) - \phi(d_2^{\mathcal{B}})\}^2$. |
| Regret Bound | $R(\hat{d}^{\mathcal{B}})$ | $\mathbb{E}\{Y(d^{\mathcal{B}*}(X))\} - \mathbb{E}\{Y(\hat{d}^{\mathcal{B}}(X))\}$. |
| Decision Tree Class | $\mathcal{D}_{\text{tree}}^{\mathcal{B}}$ | Set of all depth-$D$ trees. |

decision-making. This separation highlights the robustness and flexibility of our procedure, ensuring the resulting policy is both interpretable and practical for implementation.

We provide the $\sqrt{n}$-regret bound for CAIPWL, relying on the rate doubly robust model, policy class complexity, and bounded outcome and covariate assumptions. Table 1 is a summary of definitions relevant to the theoretical result.

**Assumption 3.13** (Rate doubly robust model assumption). For any $b \in \mathcal{B}$, $l = 1, \ldots, L$, $\hat{\pi}_b^{-l}(X)$ and $\hat{\mu}_b^{-l}(X)$ satisfy:

$$\mathbb{E}\left\{\hat{\pi}_b^{-l}(X) - \pi_b(X)\right\}^2 \to 0, \ \mathbb{E}\left\{\hat{\mu}_b^{-l}(X) - \mu_b(X)\right\}^2 \to 0,$$

$$\mathbb{E}\left\{\hat{\pi}_b^{-l}(X) - \pi_b(X)\right\}^2 \mathbb{E}\left\{\hat{\mu}_b^{-l}(X) - \mu_b(X)\right\}^2 = o(n^{-1}).$$

**Assumption 3.14** (Complexity of the policy class). $\forall 0 < \epsilon < 1$, $N_H(\epsilon^2, \mathcal{D}^{\mathcal{B}}) \leq C_1 \exp(C_2 \epsilon^{-\omega})$ for some constants $C_1, C_2 > 0$, $0 < \omega < 0.5$.

**Assumption 3.15** (Bounded outcome and covariate). For all $a \in \mathcal{A}$, $Y(a)$ is bounded, and $X$ is bounded.

*Remark* 3.16. Assumption 3.13 is weaker than the standard doubly robust model assumption, which requires either the estimator of $\pi_b(X)$ or $\mu_b(X)$ to be $\sqrt{n}$-consistent. Instead, Assumption 3.13 permits a trade-off between their accuracies, requiring only that the product of their error terms scales as $o(n^{-1})$. Modern machine learning methods offer effective estimators for these quantities. Assumption 3.14 requires the logarithm of the policy class covering number to grow at a low-order polynomial rate with $1/\epsilon$, a condition satisfied by the finite-depth trees considered here (Zhou et al., 2023). Assumption 3.15, required only for the results in Section 3.3, is a standard regularity condition in the policy learning literature.

**Proposition 3.17** (Regret bound of CAIPWL). *Under Assumptions 2.1, 3.13, 3.14, and 3.15, suppose that $\mu_a(X) =*

$\mu_{a'}(X)$ for all $\delta(a) = \delta(a')$. For the $\hat{d}^{\mathcal{B}}$ learned from Algorithm 2, we have $R(\hat{d}^{\mathcal{B}}) = O_{\mathbb{P}}\left(\kappa(\mathcal{D}^{\mathcal{B}}) \sqrt{V_*/n}\right)$.

**Proposition 3.18** (Regret bound of policy tree). *Under Assumptions 2.1, 3.13, and 3.15, suppose that $\mu_a(X) = \mu_{a'}(X)$ for all $\delta(a) = \delta(a')$. For the $\hat{d}^{\mathcal{B}}$ learned from Algorithm 2 with $\mathcal{D}^{\mathcal{B}} = \mathcal{D}_{\text{tree}}^{\mathcal{B}}$, we have $R(\hat{d}^{\mathcal{B}}) = O_{\mathbb{P}}\left(\left\{\sqrt{(2^D - 1)\log p + 2^D \log M} + \frac{4}{3} D^{\frac{1}{4}} \sqrt{2^D - 1}\right\} \sqrt{V_*/n}\right)$.*

## 4. Numerical Experiments

### 4.1. Synthetic Scenarios

We considered $M = 4$ treatment groups, with the structure summarized in Table 2. Each group comprises $|\mathcal{G}_b^*| = 4$ treatments sharing identical outcome mean functions, as detailed in Table 4. We considered covariate distributions with shifts and varying sample sizes across treatment groups, as shown in Table 3. The covariance matrices are:

$$\Sigma_1 = \begin{pmatrix} 1 & -0.25 \\ -0.25 & 1 \end{pmatrix}, \quad \Sigma_2 = \begin{pmatrix} 1 & -0.3 \\ -0.3 & 1 \end{pmatrix}.$$

*Table 2.* Group structure.

| Group | 1 | 2 | 3 | 4 |
|---|---|---|---|---|
| Treatment | {1,2,3,4} | {5,6,7,8} | {9,10,11,12} | {13,14,15,16} |

As a baseline, we implemented the CAIPWL method (Zhou et al., 2023) without calibration weighting or fusion, using the default tuning in the R package `policytree` to learn a depth-3 policy tree. We further implemented the fusion step both with and without calibration weighting, using CAIPWL to learn the corresponding optimal policy trees. In the fusion step, fused lasso uses extended BIC (Chen & Chen, 2008)

*Table 3.* Covariate distribution.

| Treatment | Covariate | Sample Size |
|---|---|---|
| {1,5,9,13} | $X_1 \sim Bernoulli(0.3)$
$(X_2, X_3)^T \| X_1 = 1 \sim N((1, -1)^T, \Sigma_1)$
$(X_2, X_3)^T \| X_1 = 0 \sim N((-1, 1)^T, \Sigma_2)$ | 150 |
| {2,6,10,14} | $X_1 \sim Bernoulli(0.4)$
$(X_2, X_3)^T \| X_1 = 1 \sim N((1, -1)^T, \Sigma_1)$
$(X_2, X_3)^T \| X_1 = 0 \sim N((-1, 1)^T, \Sigma_2)$ | 125 |
| {3,7,11,15} | $X_1 \sim Bernoulli(0.5)$
$(X_2, X_3)^T \| X_1 = 1 \sim N((1, -1)^T, \Sigma_1)$
$(X_2, X_3)^T \| X_1 = 0 \sim N((-1, 1)^T, \Sigma_2)$ | 100 |
| {4,8,12,16} | $X_1 \sim Bernoulli(0.6)$
$(X_2, X_3)^T \| X_1 = 1 \sim N((1, -1)^T, \Sigma_1)$
$(X_2, X_3)^T \| X_1 = 0 \sim N((-1, 1)^T, \Sigma_2)$ | 75 |

*Table 4.* Outcome mean functions for each group.

| $b$ | $\mu_b(X)$ |
|---|---|
| 1 | $3\exp\{0.7 + 0.1X_1 - 0.3X_2 - 0.2X_3^2 + 0.4\text{sign}(X_2^2 + 3X_3 - 2.5)\}$ |
| 2 | $3\exp\{0.5 + 0.1X_1 + 0.15X_2 - 0.3X_3^2 + 0.5\text{sign}(X_2^2 + 3X_3 - 2.5)\}$ |
| 3 | $3\exp\{0.6 + 0.1X_1 - 0.15X_2 - 0.3X_3 + 0.6\text{sign}(X_2^2 + 3X_3 - 2.5)\}$ |
| 4 | $3\exp\{0.6 + 0.1X_1 + 0.2X_2 - 0.1X_3 - 0.1X_3^2 + 0.7\text{sign}(X_2^2 - X_3 - 2)\}$ |

for model selection. Treatments are grouped if the Euclidean distance between their fused lasso estimates is less than 0.25. Additionally, for comparison, we implemented the method proposed by Ma et al. (2022), where the group structure is learned using fused Lasso without calibration weighting, and linear working models are used for both treatment fusion and policy learning.

We used the adjusted Rand index (ARI) (Gates & Ahn, 2017) to assess the quality of the fusion by comparing it to the true underlying group structure presented in Table 2. The ARI measures the similarity between two clusterings while accounting for random chance. It evaluates how pairs of items are grouped together or separated in both clusterings, adjusting for the possibility that some agreement could occur by chance. ARI produces a score between -1 and 1, where 1 indicates perfect agreement, 0 suggests no better than random chance, and negative values indicate worse-than-random fusion. We then generated a large dataset following the same distribution as the sample dataset and used the average outcome mean function over the entire dataset as the test value. We conducted 200 replications of the learning-testing procedure, with the average results summarized in Table 5.

*Table 5.* Simulation results for $K = 16$.

| Method | ARI | Number of groups | Value |
|---|---|---|---|
| policy tree (baseline) | / | 16 | 8.77 (0.08) |
| fusion + policy tree | 0.26 (0.14) | 10.725 (1.93) | 8.78 (0.09) |
| CW + fusion + policy tree | **0.96** (0.06) | **4.335** (0.60) | **8.89** (0.11) |
| Ma et al. (2022) | 0.26 (0.14) | 10.725 (1.93) | 8.51 (0.12) |

CW = Calibration Weighting. ARI (Adjusted Rand Index for fusion quality) and policy value: higher is better. Oracle number of groups = 4. Numbers in parentheses are Monte Carlo standard errors. Results are averaged over 200 runs.

We observed that, due to heterogeneity in the covariate distribution across different treatment groups, fusion without calibration weighting resulted in an average ARI of only

0.26, whereas fusion with calibration weighting achieved a significantly higher average ARI of 0.96. For the "fusion + policy tree" approach, the poor quality of fusion led to a lower average testing value compared to the "CW + fusion + policy tree" method. However, both approaches still outperformed the baseline. In contrast, the method proposed by Ma et al. (2022) suffered from misspecified outcome mean functions, yielding results that were even worse than the baseline. We perform additional simulations with increased $K$ and fixed $n$, and under a misspecified weighting model, deferring the details to Section B.

### 4.2. Real Data Application

We illustrate the proposed methods through an application to data of patients with Chronic Lymphocytic Leukemia (CLL) and Small Lymphocytic Lymphoma (SLL) from the nationwide Flatiron Health electronic health record-derived database. The Flatiron Health database is a longitudinal database, comprising de-identified patient-level structured and unstructured data, curated via technology-enabled abstraction (Ma et al., 2020; Birnbaum et al., 2020). During the study period, the de-identified data originated from approximately 280 US cancer clinics (∼800 sites of care; primarily community oncology settings). The data are de-identified and subject to obligations to prevent re-identification and protect patient confidentiality. CLL and SLL are slow-growing, indolent hematologic malignancies that primarily affect lymphocytes. In CLL, cancer cells are mainly found in the blood and bone marrow, while in SLL, they are mostly located in the lymph nodes. The relative 5-year survival rate following an initial CLL diagnosis is estimated to be 88.1% (source: SEER Cancer Statistics).

The dataset includes 10,346 patients who received first line of therapy (LOT), with details provided in Table 6. The primary outcome is patient overall survival status (1 for survival, 0 for death), along with 10 covariates: race, region, PayerBin, SES Index (2015-2019), gender, ECOG score, Rai stage, lymphadenopathy, age at the start of first LOT, and the time from diagnosis to the initiation of first LOT.

*Table 6.* Sample size for each treatment.

| Treatment | Number of patients |
|---|---|
| cBTKi mono | 3392 |
| AntiCD20 + Chemotherapy Only | 1726 |
| AntiCD20 mono | 1230 |
| BCL2i + AntiCD20 Only | 463 |
| cBKTi + AntiCD20 Only | 408 |
| Chemotherapy Only | 215 |
| Other | 412 |
| Total | 10346 |

We implemented the proposed "CW + fusion + policy tree"

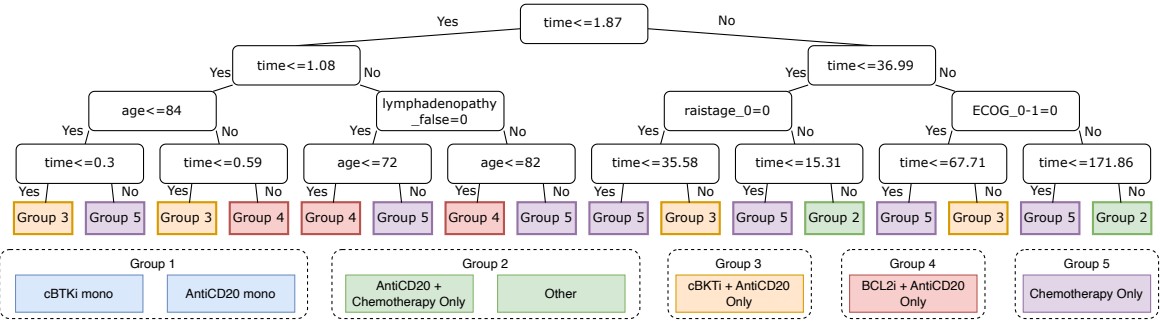

*Figure 1.* The fusion results and the learned optimal policy tree assigns patients to grouped treatments based on covariate splits. `time` is the time from diagnosis to the first LOT; `age` is the age at the start of the first LOT; `lymphadenopathy_false=1` indicates no lymph node swelling, and `lymphadenopathy_false=0` indicates its presence; `raistage_0=1` indicates Rai stage 0 (low risk), and `raistage_0=0` indicates stages I-IV (intermediate to high risk); `ECOG_0-1=1` indicates an ECOG score of 0 or 1 (good functional status), and `ECOG_0-1=0` indicates a score of 2-5 (diminished functional status).

method described in Algorithms 1 and 2. All 10 covariates were included in the calibration weighting and the estimation of nuisance functions in CAIPWL. The algorithm offers flexibility by enabling a subset of covariates to be used for learning the ITR, in contrast to methods that require the same covariates for weighting, modeling nuisance functions, and ITR learning. Certain confounders, such as race, region, and proxies of social status like PayerBin and SES Index (2015-2019), were excluded from ITR learning to avoid their use in treatment assignment. As a result, the remaining six covariates were used for fused Lasso and ITR.

The fusion results and the learned optimal policy tree are presented in Figure 1. The following insights can be drawn: (i) Two monotherapies are grouped together in Group 1, reflecting their similar mechanisms of action and treatment intensity. (ii) Combination therapies are assigned to distinct groups, while chemotherapy-only forms its own separate group. (iii) Older patients or those with shorter time since diagnosis are more likely recommended to Group 5 (chemotherapy-only), likely due to limited treatment tolerance and the need for immediate intervention. (iv) Relatively younger patients or those with longer time since diagnosis tend to be recommended to Groups 2, 3, or 4 (combination therapies), likely due to their better functional status to tolerate aggressive treatments, and longer diagnostic times often indicate a chronic disease course requiring more targeted interventions to manage progression. Our findings provide valuable insights for guiding future individualized treatment strategies while ensuring their practical feasibility.

## 5. Conclusion

This paper introduces a calibration-weighted treatment fusion procedure to address the challenges of many treatments in ITR learning. By leveraging treatment similarities and robustly balancing covariates, our method employs weighted

fused Lasso to recover the latent group structure of treatments, providing theoretical guarantees of consistency and double robustness. Practitioners can seamlessly integrate the fusion results with state-of-the-art ITR learning methods, such as policy trees, which offer $\sqrt{n}$-regret bounds, enabling flexible and interpretable decision-making. Simulation studies highlight the superiority of our method over baseline and competing approaches. Additionally, we demonstrate its practical utility through a real data application, yielding clinically relevant insights.

In extreme cases where some treatment arms have few or no observations, our method may become unstable due to the lack of information, unless additional structural assumptions, such as the combinatorial structure in Agarwal et al. (2023), are imposed. This instability underscores the inherent difficulty of the problem and suggests directions for methodological improvement. Currently, the fusion step is performed once; an alternative is to adopt an iterative procedure that alternates between treatment fusion and weight estimation to enhance stability.

Our method can be extended in several directions. First, while this study focuses on a single data source, future work could enhance the procedure by integrating multi-source data or generalizing learned ITRs to other target populations (Mo et al., 2021; Chu et al., 2023; Wu & Yang, 2023; Zhang et al., 2024; Carranza & Athey, 2024). Second, beyond ITR estimation, providing inference and uncertainty quantification is critical (Liang et al., 2022; Ghosh et al., 2023; Cheng & Yang, 2024), especially in high-stakes contexts like medicine, with conformal prediction offering a promising approach (Osama et al., 2020; Taufiq et al., 2022; Zhang et al., 2023). Finally, high-dimensional treatments also occur in heterogeneous treatment effect (HTE) estimation (Goplerud et al., 2025), and extending our procedure to HTE frameworks would enhance its applicability.

## Acknowledgment

We thank the anonymous reviewers and meta-reviewers of ICML 2025 for their valuable feedback, which led to a greatly improved manuscript. Yang was partially supported by the National Science Foundation grant SES 2242776 and the National Institutes of Health grants 1R01ES031651 and 1R01HL169347.

## Impact Statement

This paper presents work whose goal is to advance the field of policy learning. There are many potential societal consequences of our work, none which we feel must be specifically highlighted here.

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

# A. Proof

## A.1. Rationale for Assumption 3.2

For every treatment $a \in \mathcal{A}$, calibration weighting is an optimization problem and can be solved using the method of Lagrange multipliers:

$$L_a(w_1, \ldots, w_n) = \sum_{i:A_i=a} \frac{(n_a w_i)^{\gamma+1} - 1}{\gamma(\gamma+1)} - n\lambda^\top \sum_{i:A_i=a} w_i(X_i - \bar{X}) + n\varphi\left(1 - \sum_{i:A_i=a} w_i\right).$$

Minimizing $L_a(w_1, \ldots, w_n)$ gives:

$$\hat{w}_i = w(X_i; \hat{\lambda}) = \frac{\rho_\gamma[\hat{\lambda}^\top(X_i - \bar{X})]}{\sum_{j:A_j=a} \rho_\gamma[\hat{\lambda}^\top(X_j - \bar{X})]},$$

where the function $\rho_\gamma(x)$ for different $\gamma$ values are summarized in Table 7, and $\hat{\lambda}$ solves the equation

$$\sum_{i:A_i=a} \rho_\gamma[\lambda^\top(X_i - \bar{X})](X_i - \bar{X}) = 0.$$

Therefore, $\hat{\lambda}$ is an M-estimator and, under standard regularity conditions for M-estimators (Boos & Stefanski, 2013), it is root-n consistent.

*Table 7.* $\rho_\gamma(x)$ for Cressie-Read family.

| $\gamma$ | $h_\gamma(w)$ | $\rho_\gamma(x)$ |
|---|---|---|
| $-1$ | $-\ln(nw)$ | $(1-x)^{-1}$ |
| $0$ | $nw\ln(nw)$ | $\exp(x)$ |
| $\gamma$ | $\dfrac{(nw)^{\gamma+1} - 1}{\gamma(\gamma+1)}$ | $(1 + \gamma x)^{1/\gamma}$ |

## A.2. Lemma A.1

**Lemma A.1.** *Suppose Assumptions 2.1 and 3.3 hold. For any $i = 1, \ldots, n$, $j = 1, \ldots, p$, and $a \in \mathcal{A}$,*

$$\mathbb{E}\{\mathbb{I}(A_i = a)w_i^* X_{ij}\varepsilon_i(a)\} = 0.$$

*Proof of Lemma A.1.* By the unconfoundedness in Assumption 2.1, we have

$$\mathbb{E}\{\mathbb{I}(A_i = a)w_i^* X_{ij}\varepsilon_i(a)\} = \mathbb{E}\big[\mathbb{E}\{\mathbb{I}(A_i = a)w_i^* X_{ij}\varepsilon_i(a) \mid X_i\}\big]$$
$$= \mathbb{E}\big[\mathbb{E}\{\mathbb{I}(A_i = a)w_i^* \mid X_i\}X_{ij}\mathbb{E}\{\varepsilon_i(a) \mid X_i\}\big].$$

If (i) (correct calibration weighting) in Assumption 3.3 holds, i.e., $\mathbb{E}\{\mathbb{I}(A_i = a)w_i^* \mid X_i\} = 1$, we have

$$\mathbb{E}\big[\mathbb{E}\{\mathbb{I}(A_i = a)w_i^* \mid X_i\}X_{ij}\mathbb{E}\{\varepsilon_i(a) \mid X_i\}\big] = \mathbb{E}\big[X_{ij}\mathbb{E}\{\varepsilon_i(a) \mid X_i\}\big] = \mathbb{E}\big[X_{ij}\varepsilon_i(a)\big].$$

By the definitions of the projection vector and projection residual:

$$\zeta_a^* := \operatorname*{argmin}_{\zeta \in \mathbb{R}^p} \mathbb{E}\left\{\tilde{Y}(a) - X^\top \zeta\right\}^2, \quad \text{and} \quad \varepsilon(a) := \tilde{Y}(a) - X^\top \zeta_a^*, \tag{12}$$

the result follows from

$$E\left\{X^\top \varepsilon(a)\right\} = E\left[X^\top\left\{\tilde{Y}(a) - X^\top \zeta_a^*\right\}\right] = 0.$$

Note that $E\left\{X^\top \varepsilon(a)\right\} = 0$ does not require a linear model between $\tilde{Y}(a)$ and $X$; it holds solely due to the projection (12).

If (ii) (correct outcome model) in Assumption 3.3 holds, i.e., $\mathbb{E}\{\varepsilon_i(a) \mid X_i\} = 0$, we have

$$\mathbb{E}\big[\mathbb{E}\{\mathbb{I}(A_i = a)w_i^* \mid X_i\}X_{ij}\mathbb{E}\{\varepsilon_i(a) \mid X_i\}\big] = 0.$$

Note that the above equation does not require calibration weighting to be correct; it holds solely due to $E\{\varepsilon(a) \mid X\} = 0$, a condition stronger than $E\{X^\top \varepsilon(a)\} = 0$. This condition is also referred to as the exogeneity assumption or the zero conditional mean condition (Wooldridge, 2012). $\qquad\square$

### A.3. Proof of Theorem 3.8

*Proof of Theorem 3.8.* We rewrite $\widehat{\boldsymbol{\beta}}_b$ to relate it to the potential outcomes

$$\widehat{\boldsymbol{\beta}}_b = \min_{\boldsymbol{\beta}_m \in \mathbb{R}^p} \frac{1}{2n} \sum_{i=1}^n \sum_{a \in \mathcal{G}_b^*} \mathbb{I}(A_i = a)\widehat{w}_i\left(\tilde{Y}_i - X_i^\top \boldsymbol{\beta}_m\right)^2$$

$$= \min_{\boldsymbol{\beta}_m \in \mathbb{R}^p} \frac{1}{2n} \sum_{i=1}^n \sum_{a \in \mathcal{G}_b^*} \mathbb{I}(A_i = a)\widehat{w}_i\left(\tilde{Y}_i(a) - X_i^\top \boldsymbol{\beta}_m\right)^2$$

$$= \min_{\boldsymbol{\beta}_m \in \mathbb{R}^p} \frac{1}{2n} \sum_{i=1}^n \sum_{a \in \mathcal{G}_b^*} \mathbb{I}(A_i = a)\widehat{w}_i\left(X^\top \boldsymbol{\beta}_b^* + \varepsilon(a) - X_i^\top \boldsymbol{\beta}_m\right)^2.$$

where the second equation is due to $\mathbb{I}(A_i = a)\tilde{Y}_i = \mathbb{I}(A_i = a)\tilde{Y}_i(a)$ and the third equation is due to (10). Then, the least squares estimation leads to

$$\widehat{\boldsymbol{\beta}}_b - \boldsymbol{\beta}_b^* = \underbrace{\left\{\sum_{i=1}^n \sum_{a \in \mathcal{G}_b^*} \mathbb{I}(A_i = a)\widehat{w}_i X_i X_i^\top\right\}^{-1}}_{\Gamma_{\widehat{w}X[b]}} \underbrace{\left\{\sum_{i=1}^n \sum_{a \in \mathcal{G}_b^*} \mathbb{I}(A_i = a)\widehat{w}_i X_i \varepsilon_i(a)\right\}}_{\Gamma_{\widehat{w}\varepsilon[b]}}.$$

We have

$$\max_{b \in \mathcal{B}} \|\widehat{\boldsymbol{\beta}}_b - \boldsymbol{\beta}_b^*\|_\infty \leq \max_{b \in \mathcal{B}} \|\Gamma_{\widehat{w}X[b]}^{-1}\|_\infty \|\Gamma_{\widehat{w}\varepsilon[b]}\|_\infty. \tag{13}$$

We examine $\|\Gamma_{\widehat{w}X[b]}^{-1}\|_\infty$ and $\|\Gamma_{\widehat{w}\varepsilon[b]}\|_\infty$, respectively.

**Step 1** (Bound $\|\Gamma_{\widehat{w}X[b]}^{-1}\|_\infty$). By Assumption 3.2, $\exists \epsilon > 0$, with probability at least $1 - \iota_n$ (where $\iota_n \to 0$ as $n \to \infty$), $\widehat{w}_i \geq w_i^* - |\widehat{w}_i - w_i^*| \geq C_1 - \epsilon := C_1'$, for any $i$. Thus, we have

$$\|\Gamma_{\widehat{w}X[b]}^{-1}\|_2 = \Lambda_{\max}(\Gamma_{\widehat{w}X[b]}^{-1}) = \frac{1}{\Lambda_{\min}(\Gamma_{\widehat{w}X[b]})} \leq \frac{1}{C_4 C_1' N_{\min}},$$

where the last inequality is due to Assumptions 3.5. Then, we have

$$\max_{b \in \mathcal{B}} \|\Gamma_{\widehat{w}X[b]}^{-1}\|_\infty \leq \sqrt{p} \max_{b \in \mathcal{B}} \|\Gamma_{\widehat{w}X[b]}^{-1}\|_2 \leq \frac{\sqrt{p}}{C_4 C_1' N_{\min}}. \tag{14}$$

**Step 2** (Bound $\|\Gamma_{\widehat{w}\varepsilon[b]}\|_\infty$). We first bound

$$\Gamma_{w\varepsilon[b]} := \sum_{i=1}^n \sum_{a \in \mathcal{G}_b^*} \mathbb{I}(A_i = a)w_i^* X_i \varepsilon_i(a).$$

By Assumptions 3.2 and 3.5, we have

$$\sum_{i=1}^n \sum_{a \in \mathcal{G}_b^*} \mathbb{I}(A_i = a)(w_i^* X_{ij})^2 \leq nC_2^2 C_3.$$

Combined with Lemma A.1 and Assumption 3.6, for any $j = 1, \ldots, p$ and $b \in \mathcal{B}$, for $t > 0$, we have

$$\mathbb{P}\left(\left|\sum_{i=1}^{n} \sum_{a \in \mathcal{G}_b^*} \mathbb{I}(A_i = a) w_i^* X_{ij} \varepsilon_i(a)\right| > \sqrt{n C_2^2 C_3} t\right) \leq 2 \exp(-t^2/2\sigma_\varepsilon^2).$$

Then, we have

$$\mathbb{P}\left(\max_{b \in \mathcal{B}} \|\Gamma_{w\varepsilon[b]}\|_\infty > \sqrt{n C_2^2 C_3} t\right) \leq 2Mp \exp(-t^2/2\sigma_\varepsilon^2).$$

Letting $2Mp \exp(-t^2/2\sigma_\varepsilon^2) = 2Mp/n$, we have that with probability at least $1 - 2Mp/n$,

$$\max_{b \in \mathcal{B}} \|\Gamma_{w\varepsilon[b]}\|_\infty \leq \sqrt{2C_2^2 C_3} \sigma_\varepsilon \sqrt{n \log(n)}. \tag{15}$$

Finally, by Assumption 3.2,

$$\max_{b \in \mathcal{B}} \|\Gamma_{\hat{w}X[b]}^{-1}\|_\infty \|\Gamma_{\hat{w}\varepsilon[b]} - \Gamma_{w\varepsilon[b]}\|_\infty = O_{\mathbb{P}}\left(\frac{\sqrt{pn}}{N_{\min}}\right). \tag{16}$$

The result follows from (13), (14), (15), and (16). $\qquad\square$

### A.4. Proof of Theorem 3.12

*Proof of Theorem 3.12.* Recall that the true group structure is $\cup_{b=1}^{M} \mathcal{G}_b^*$. The space of $\zeta$'s that have the true group structure is defined as

$$\mathcal{Z}^{\mathrm{or}} := \left\{\zeta = (\zeta_1^\top, \ldots, \zeta_K^\top)^\top \in \mathbb{R}^{Kp} : \forall b \in \mathcal{B}, \forall a, a' \in \mathcal{G}_b^*, \zeta_a = \zeta_{a'}\right\}.$$

Note that $\hat{\zeta}^{\mathrm{or}} \in \mathcal{Z}^{\mathrm{or}}$. Define the mapping

$$T : \mathbb{R}^{Kp} \to \mathcal{Z}^{\mathrm{or}}, \quad \zeta \mapsto \bar{\zeta},$$

where $\bar{\zeta} = (\bar{\zeta}_1^\top, \ldots, \bar{\zeta}_K^\top)^\top$ and $\bar{\zeta}_a = \sum_{a' \in \mathcal{G}_b^*} \zeta_{a'} / |\mathcal{G}_b^*|, \forall a \in \mathcal{G}_b^*$. Define the neighbor of projection vector $\zeta^*$ as

$$\Theta = \left\{\zeta \in \mathbb{R}^{Kp} : \|\zeta - \zeta^*\|_\infty \leq \phi_n\right\}.$$

Note that $\forall \zeta \in \Theta$, we have $T(\zeta) \in \Theta$. Define the event $E_1 = \{\hat{\zeta}^{\mathrm{or}} \in \Theta\}$. By Theorem 3.8, we have $\mathbb{P}(E_1) \geq 1 - 2Mp/n - \iota_n$. The result follows from the following two statements, each of which we will prove separately.

- **Statement 1** On event $E_1$, for all $\zeta \in \Theta$ such that $T(\zeta) \neq \hat{\zeta}^{\mathrm{or}}$, we have $Q_n(T(\zeta)) > Q_n(\hat{\zeta}^{\mathrm{or}})$.

- **Statement 2** There exists an event $E_2$ such that $\mathbb{P}(E_2) \geq 1 - Kp/n - \iota_n$, and on $E_2$, for all $\zeta \in \Theta$, we have $Q_n(\zeta) \geq Q_n(T(\zeta))$.

**Proof of Statement 1.** We examine $L_n(T(\zeta)) - L_n(\hat{\zeta}^{\mathrm{or}})$ and $P_n(T(\zeta)) - P_n(\hat{\zeta}^{\mathrm{or}})$, respectively.

**Step 1.1** (Examine $L_n$). By definition, restricted to $\mathcal{Z}^{\mathrm{or}}$, $\hat{\zeta}^{\mathrm{or}}$ is the unique minimizer of $L_n(\zeta)$, that is, for all $\zeta \in \Theta$ such that $T(\zeta) \neq \hat{\zeta}^{\mathrm{or}}$,

$$L_n(T(\zeta)) > L_n(\hat{\zeta}^{\mathrm{or}}). \tag{17}$$

**Step 1.2** (Examine $P_n$). For any $\bar{\zeta} = (\bar{\zeta}_1^\top, \ldots, \bar{\zeta}_K^\top)^\top \in \mathcal{Z}^{\mathrm{or}} \cap \Theta$ (including the case $\bar{\zeta} = \hat{\zeta}^{\mathrm{or}}$),

$$P_n(\bar{\zeta}) = \sum_{1 \leq a < a' \leq K} p_{\lambda_n}\left(\|\bar{\zeta}_a - \bar{\zeta}_{a'}\|_1\right).$$

If treatments belong to different groups, i.e., $a \in \mathcal{G}_b^*$ and $a' \in \mathcal{G}_{b'}^*$ with $b \neq b'$, by Assumption 3.10, we have

$$\begin{aligned}
\|\bar{\zeta}_a - \bar{\zeta}_{a'}\|_1 &\geq \|\bar{\zeta}_a - \bar{\zeta}_{a'}\|_\infty \\
&\geq \|\zeta_a^* - \zeta_{a'}^*\|_\infty - 2\|\bar{\zeta} - \zeta^*\|_\infty \\
&\geq c\lambda_n - 2\phi_n \\
&\gg c\lambda_n/2.
\end{aligned}$$

Then, $p_{\lambda_n}\left(\left\|\bar{\boldsymbol{\zeta}}_a - \bar{\boldsymbol{\zeta}}_{a'}\right\|_1\right)$ is a constant by Assumption 3.10. If treatments belong to the same group, i.e., $a, a' \in \mathcal{G}_b^*$, we have $\left\|\bar{\boldsymbol{\zeta}}_a - \bar{\boldsymbol{\zeta}}_{a'}\right\|_1 = 0$, thus $p_{\lambda_n}\left(\left\|\bar{\boldsymbol{\zeta}}_a - \bar{\boldsymbol{\zeta}}_{a'}\right\|_1\right) = 0$. Overall, we have that $P_n(\bar{\boldsymbol{\zeta}})$ is a constant. Since $T(\boldsymbol{\zeta}) \in \mathcal{Z}^{\mathrm{or}} \cap \Theta$ and $\widehat{\boldsymbol{\zeta}}^{\mathrm{or}} \in \mathcal{Z}^{\mathrm{or}} \cap \Theta$ on $E_1$, we have

$$P_n(T(\boldsymbol{\zeta})) = P_n(\widehat{\boldsymbol{\zeta}}^{\mathrm{or}}). \tag{18}$$

The Statement 1 follows from (17) and (18).

**Proof of Statement 2**: Let $\bar{\boldsymbol{\zeta}} := T(\boldsymbol{\zeta})$. We examine $L_n(\boldsymbol{\zeta}) - L_n(\bar{\boldsymbol{\zeta}})$ and $P_n(\boldsymbol{\zeta}) - P_n(\bar{\boldsymbol{\zeta}})$, respectively.

**Step 2.1** (Examine $L_n$). By Taylor expansion, there is $0 < \xi < 1$ such that $\tilde{\boldsymbol{\zeta}} = \xi\boldsymbol{\zeta} + (1 - \xi)\bar{\boldsymbol{\zeta}} \in \Theta$, we have

$$
\begin{aligned}
L_n(\boldsymbol{\zeta}) - L_n(\bar{\boldsymbol{\zeta}}) &= -\frac{1}{n}\sum_{i=1}^n \sum_{a \in \mathcal{A}} \mathbb{I}(A_i = a)\widehat{w}_i\left(\tilde{Y}_i - X_i^\top\tilde{\boldsymbol{\zeta}}_a\right)\left(X_i^\top\boldsymbol{\zeta}_a - X_i^\top\bar{\boldsymbol{\zeta}}_a\right) \\
&= -\frac{1}{n}\sum_{i=1}^n \sum_{a \in \mathcal{A}} \mathbb{I}(A_i = a)\widehat{w}_i\left\{\tilde{Y}_i(a) - X_i^\top\tilde{\boldsymbol{\zeta}}_a\right\}\left(X_i^\top\boldsymbol{\zeta}_a - X_i^\top\bar{\boldsymbol{\zeta}}_a\right) \\
&= -\frac{1}{n}\sum_{i=1}^n \sum_{a \in \mathcal{A}} \mathbb{I}(A_i = a)\widehat{w}_i\left\{X_i^\top\boldsymbol{\zeta}_a^* + \varepsilon_i(a) - X_i^\top\tilde{\boldsymbol{\zeta}}_a\right\}\left(X_i^\top\boldsymbol{\zeta}_a - X_i^\top\bar{\boldsymbol{\zeta}}_a\right) \\
&= -\frac{1}{n}\sum_{i=1}^n \sum_{a \in \mathcal{A}} \mathbb{I}(A_i = a)\widehat{w}_i\left\{X_iX_i^\top(\boldsymbol{\zeta}_a^* - \tilde{\boldsymbol{\zeta}}_a) + X_i\varepsilon_i(a)\right\}^\top\left(\boldsymbol{\zeta}_a - \bar{\boldsymbol{\zeta}}_a\right).
\end{aligned}
$$

Let

$$\boldsymbol{v}_a := \frac{1}{n}\sum_{i=1}^n \mathbb{I}(A_i = a)\widehat{w}_i\left\{X_iX_i^\top(\boldsymbol{\zeta}_a^* - \tilde{\boldsymbol{\zeta}}_a) + X_i\varepsilon_i(a)\right\}.$$

Then, we have $L_n(\boldsymbol{\zeta}) - L_n(\bar{\boldsymbol{\zeta}}) = -\sum_{a \in \mathcal{A}} \boldsymbol{v}_a^\top(\boldsymbol{\zeta}_a - \bar{\boldsymbol{\zeta}}_a)$. Since $\bar{\boldsymbol{\zeta}}_a = \sum_{a' \in \mathcal{G}_b^*} \boldsymbol{\zeta}_{a'}/|\mathcal{G}_b^*|, \forall a \in \mathcal{G}_b^*$, by some algebra, we have

$$
\begin{aligned}
L_n(\boldsymbol{\zeta}) - L_n(\bar{\boldsymbol{\zeta}}) &= -\sum_{b \in \mathcal{B}}\sum_{a \in \mathcal{G}_b^*} \boldsymbol{v}_a^\top(\boldsymbol{\zeta}_a - \bar{\boldsymbol{\zeta}}_a) \\
&= -\sum_{b \in \mathcal{B}}\sum_{a \in \mathcal{G}_b^*} \boldsymbol{v}_a^\top\left(\boldsymbol{\zeta}_a - \frac{\sum_{a' \in \mathcal{G}_b^*}\boldsymbol{\zeta}_{a'}}{|\mathcal{G}_b^*|}\right) \\
&= -\sum_{b \in \mathcal{B}}\sum_{a \in \mathcal{G}_b^*}\sum_{a' \in \mathcal{G}_b^*} \frac{\boldsymbol{v}_a^\top(\boldsymbol{\zeta}_a - \boldsymbol{\zeta}_{a'})}{|\mathcal{G}_b^*|} \\
&= -\sum_{b \in \mathcal{B}}\sum_{a \in \mathcal{G}_b^*}\sum_{a' \in \mathcal{G}_b^*} \frac{(\boldsymbol{v}_a - \boldsymbol{v}_{a'})^\top(\boldsymbol{\zeta}_a - \boldsymbol{\zeta}_{a'})}{2|\mathcal{G}_b^*|} \\
&= -\sum_{b \in \mathcal{B}}\sum_{a, a' \in \mathcal{G}_b^*, a < a'} \frac{(\boldsymbol{v}_a - \boldsymbol{v}_{a'})^\top(\boldsymbol{\zeta}_a - \boldsymbol{\zeta}_{a'})}{|\mathcal{G}_b^*|} \\
&\geq -\sum_{b \in \mathcal{B}}\sum_{a, a' \in \mathcal{G}_b^*, a < a'} \frac{\|\boldsymbol{v}_a - \boldsymbol{v}_{a'}\|_\infty \|\boldsymbol{\zeta}_a - \boldsymbol{\zeta}_{a'}\|_1}{|\mathcal{G}_b^*|}.
\end{aligned}
$$

By Assumptions 3.5 and 3.2, we have

$$
\begin{aligned}
\left\|\frac{1}{n}\sum_{i=1}^n \mathbb{I}(A_i = a)\widehat{w}_iX_iX_i^\top(\boldsymbol{\zeta}_a^* - \tilde{\boldsymbol{\zeta}}_a)\right\|_\infty &\leq \left\|\frac{1}{n}\sum_{i=1}^n \mathbb{I}(A_i = a)\widehat{w}_iX_iX_i^\top\right\|_\infty \left\|\boldsymbol{\zeta}_a^* - \tilde{\boldsymbol{\zeta}}_a\right\|_\infty \\
&= O_\mathbb{P}(p\,\phi_n).
\end{aligned}
$$

Following a similar derivation as in Step 2 of the proof of Theorem 3.8, there exists an event $E_2$ such that $\mathbb{P}(E_2) \geq 1 - Kp/n - \iota_n$, and on $E_2$, we have

$$\left\|\frac{1}{n}\sum_{i=1}^n \mathbb{I}(A_i = a)\widehat{w}_iX_i\varepsilon_i(a)\right\|_\infty = O(\sqrt{n\log(n)}).$$

Thus, we have

$$\|\boldsymbol{v}_a - \boldsymbol{v}_{a'}\|_\infty \leq 2 \max_{a \in \mathcal{A}} \|\boldsymbol{v}_a\|_\infty = O_\mathbb{P}\left(p\,\phi_n + \sqrt{n\log(n)}\right),$$

and

$$L_n(\boldsymbol{\zeta}) - L_n(\bar{\boldsymbol{\zeta}}) \geq -\sum_{b \in \mathcal{B}} \sum_{a,a' \in \mathcal{G}_b^*, a<a'} \frac{O_\mathbb{P}\left(p\,\phi_n + \sqrt{n\log(n)}\right)}{|\mathcal{G}_b^*|} \|\boldsymbol{\zeta}_a - \boldsymbol{\zeta}_{a'}\|_1. \tag{19}$$

**Step 2.2** (Examine $P_n$). Following similar arguments as in Step 1.2 of the proof of Statement 1, if treatments belong to different groups, i.e., $a \in \mathcal{G}_b^*$ and $a' \in \mathcal{G}_{b'}^*$ with $b \neq b'$, by Assumption 3.10, we have $\|\boldsymbol{\zeta}_a - \boldsymbol{\zeta}_{a'}\|_1 \gg c\lambda_n/2$ and $\|\bar{\boldsymbol{\zeta}}_a - \bar{\boldsymbol{\zeta}}_{a'}\|_1 \gg c\lambda_n/2$, thus,

$$p_{\lambda_n}\left(\|\boldsymbol{\zeta}_a - \boldsymbol{\zeta}_{a'}\|_1\right) - p_{\lambda_n}\left(\|\bar{\boldsymbol{\zeta}}_a - \bar{\boldsymbol{\zeta}}_{a'}\|_1\right) = 0.$$

If treatments belong to the same group, i.e., $a, a' \in \mathcal{G}_b^*$, we have $\|\bar{\boldsymbol{\zeta}}_a - \bar{\boldsymbol{\zeta}}_{a'}\|_1 = 0$, thus $p_{\lambda_n}(\|\bar{\boldsymbol{\zeta}}_a - \bar{\boldsymbol{\zeta}}_{a'}\|_1) = 0$. However, since $\|\boldsymbol{\zeta}_a - \boldsymbol{\zeta}_{a'}\|_1 \neq 0$, they are the only terms contributing to $P_n(\boldsymbol{\zeta}) - P_n(\bar{\boldsymbol{\zeta}})$. Thus, we have

$$P_n(\boldsymbol{\zeta}) - P_n(\bar{\boldsymbol{\zeta}}) = \sum_{b \in \mathcal{B}} \sum_{a,a' \in \mathcal{G}_b^*, a<a'} p_{\lambda_n}\left(\|\boldsymbol{\zeta}_a - \boldsymbol{\zeta}_{a'}\|_1\right)$$

$$= \sum_{b \in \mathcal{B}} \sum_{a,a' \in \mathcal{G}_b^*, a<a'} \frac{p_{\lambda_n}\left(\|\boldsymbol{\zeta}_a - \boldsymbol{\zeta}_{a'}\|_1\right)}{\|\boldsymbol{\zeta}_a - \boldsymbol{\zeta}_{a'}\|_1} \|\boldsymbol{\zeta}_a - \boldsymbol{\zeta}_{a'}\|_1. \tag{20}$$

We have $\|\boldsymbol{\zeta}_a - \boldsymbol{\zeta}_{a'}\|_1 \leq \|\boldsymbol{\zeta}_a - \boldsymbol{\zeta}_a^*\|_1 + \|\boldsymbol{\zeta}_{a'} - \boldsymbol{\zeta}_{a'}^*\|_1 \leq 2\phi_n \to 0$. By Assumption 3.10, that is, $p_{\lambda_n}(\cdot) = \lambda_n \rho(\cdot)$, $\rho'(0+) = 1$, and $\lambda_n \gg p\,\phi_n/K_{\min} + \sqrt{n\log(n)}/K_{\min}$, we have

$$\frac{p_{\lambda_n}\left(\|\boldsymbol{\zeta}_a - \boldsymbol{\zeta}_{a'}\|_1\right)}{\|\boldsymbol{\zeta}_a - \boldsymbol{\zeta}_{a'}\|_1} \geq O\left(\frac{p\,\phi_n + \sqrt{n\log(n)}}{K_{\min}}\right). \tag{21}$$

The Statement 2 follows from (19) (20), and (21). $\qquad\square$

## B. Additional Simulation Results

### B.1. Simulations for increasing $K$ and fixed $n$

We keep the sample size fixed at $n = 1800$ and increase the number of treatments $K$ from 16 to 32 and 48. In such regime, our proposed method outperforms other baselines in terms of fusion quality and policy value (see Tables 8 and 9). The number of recovered groups increases slightly, as expected.

*Table 8.* Simulation results for $K = 32$

| Method | ARI | Number of groups | Value |
|---|---|---|---|
| policy tree (baseline) | / | 32 | 8.66 (0.18) |
| fusion + policy tree | 0.21 (0.09) | 16.74 (3.64) | 8.63 (0.18) |
| CW + fusion + policy tree | **0.85** (0.10) | **5.68** (1.72) | **8.80** (0.21) |
| Ma et al. (2022) | 0.21 (0.09) | 16.74 (3.64) | 8.52 (0.12) |

CW = Calibration Weighting. ARI (Adjusted Rand Index for fusion quality) and policy value: higher is better. Oracle number of groups = 4. Numbers in parentheses are Monte Carlo standard errors. Results are averaged over 200 runs.

### B.2. Simulations under misspecified weighting model

We additionally considered a scenario where the outcome mean functions are linear (i.e., correctly specified), but the weighting model is misspecified. Specifically, in calibration weighting, we excluded $X_1$ and used only $X_2$ and $X_3$. The

*Table 9.* Simulation results for $K = 48$

| Method | ARI | Number of groups | Value |
|---|---|---|---|
| policy tree (baseline) | / | 48 | 8.49 (0.20) |
| fusion + policy tree | 0.16 (0.08) | 21.00 (5.58) | 8.40 (0.30) |
| CW + fusion + policy tree | **0.74** (0.10) | **7.35** (2.31) | **8.52** (0.23) |
| Ma et al. (2022) | 0.16 (0.08) | 21.00 (5.58) | 8.41 (0.12) |

CW = Calibration Weighting. ARI (Adjusted Rand Index for fusion quality) and policy value: higher is better. Oracle number of groups = 4. Numbers in parentheses are Monte Carlo standard errors. Results are averaged over 200 runs.

outcome mean functions were set as follows:

$$Y_1 = 2.5 + 0.5X_1 - 1.5X_2 - X_3,$$
$$Y_2 = X_1 - 2X_2 - 2.5X_3,$$
$$Y_3 = 2 - 0.5X_1 + 2X_2 - 2X_3,$$
$$Y_4 = -1 + X_1 - X_2 + X_3.$$

*Table 10.* Simulation results under misspecified weighting model

| Method | ARI | Number of groups | Value |
|---|---|---|---|
| policy tree (baseline) | / | 16 | 6.35 (0.06) |
| fusion + policy tree | 0.88 (0.13) | 5.42 (1.42) | 6.41 (0.04) |
| CW + fusion + policy tree | **0.96** (0.06) | **4.46** (0.66) | **6.43** (0.02) |
| Ma et al. (2022) | 0.88 (0.13) | 5.42 (1.42) | 6.39 (0.00) |

CW = Calibration Weighting. ARI (Adjusted Rand Index for fusion quality) and policy value: higher is better. Oracle number of groups = 4. Numbers in parentheses are Monte Carlo standard errors. Results are averaged over 200 runs.

Table 10 presents the results. As the outcome models are correctly specified, both the fusion (based on a linear model) and the CW + fusion approaches achieved strong ARI scores, illustrating the double robustness property of CW + fusion. Ma et al. (2022) also performed well, as their method relies on the same linearity assumption, which holds in this setting. Nevertheless, our proposed method consistently achieved the best overall performance. Note that the simulation in Section 4.1 already showed the advantage of the proposed method when the outcome model is misspecified while the weighting model is correct.

