# OpenReview forum: "Doubly Robust Fusion of Many Treatments for Policy Learning"
_ICML.cc/2025/Conference — ICML 2025 poster_

### Official Review · Reviewer_7HRd · 2025-03-03

**Overall Recommendation:** 3

**Summary:**

The paper proposes a new methodology for policy learning in scenarios where many treatments are available. In such settings, naive methods may fail due to e.g., lack of overlap. The proposed methodology assumes that treatments can be assigned into "groups" that have the same treatment effect and learn this grouping data-driven. Then, standard policy learning is applied on top of the reduced treatment space.

**Claims And Evidence:**

Yes and no. Both theoretical and empirical evidence are provided. However, the theory relies on strong assumptions, and the experimental section could be improved (more details below).

**Essential References Not Discussed:**

I am not aware of any existing paper that proposes to learn a group treatment structure.

**Experimental Designs Or Analyses:**

The experimental section is relatively weak. The proposed method is only benchmarked on a single simulated dataset. The insights on the real-world dataset are nice but do not compare baselines/ ablations (which I am fine with since evaluation on real-world data is difficult). Most tables in Sec. 4 do not show results but details regarding experimental setups (which could also be moved to the Appendix). Furthermore, many details are not reported.
- Over how many runs are averaged? There is no variance reported in Table 5
- How are hyperparameters chosen? Did the authors use hyperparameter tuning?
- How were the lasso parameters chosen? The proposed method seems to find the correct sparseness structure, but could this be due to a different lasso penalty parameter?
- How sensitive is the proposed method with respect to violations of assumptions (e.g., what if there is no sparse group structure in the DGP?)

In summary, I think the experimental section needs to be significantly improved.

**Methods And Evaluation Criteria:**

The authors choose reasonable metrics and datasets for evaluation. Benchmarking on synthetic datasets is standard in causal inference as ground-truth is not available for real-world data.

**Other Comments Or Suggestions:**

No further comments

**Other Strengths And Weaknesses:**

Strenghts:
- The general idea to "group" multiple treatments into a single one and then apply policy learning on a reduced treatment space seems appealing, especially in settings with many treatments in which we often do not have sufficient overlap to do proper policy learning.

Weaknesses:
- The writing could be improved, especially in Sec. 3.2. The authors start introducing various quantities to outline a sketch of proof for their theoretical results. In my opinion, it would help the flow of the paper to first state the main results and then potentially provide some intuition later on.
- The proposed methodologies relies on strong assumptions. First, parametric structure is imposed on the DGP (linear interactions between treatments and covariates). Second-

**Questions For Authors:**

In summary, I like the main of idea of the paper. My two major concerns are:
- the experimental section (see above)
- The assumptions imposed and violations thereof. Could the authors elaborate on this point and outline what would happen if, e.g., the proposed method would try to enforce a group structure that does not exist within the DGP?

Another question: I am not sure if I understand the (rather complicated) motivation of using calibration weights and a linear lasso approach to learn the group structure. Why not simply train, e.g., a neural net to learn a group structure and a policy that maximizes policy value (in an end-to-end manner)?

Finally, another well-established approach to deal with large treatment spaces is to only focus on the treatments for which sufficient data is available (e.g., overlap regions; see an example here: https://proceedings.neurips.cc/paper_files/paper/2023/hash/d69103d7895f4e2083f24b664003d386-Abstract-Conference.html). How does the proposed approach compare with such methods?

**Relation To Broader Scientific Literature:**

The key contribution to learn a sparse group structure to reduce the treatment space for policy learning seems to be novel. However, there are a few related ideas in the literature on "complex" treatments that have not been acknowledged. Examples:
https://proceedings.neurips.cc/paper/2021/file/d02e9bdc27a894e882fa0c9055c99722-Paper.pdf
https://proceedings.neurips.cc/paper_files/paper/2023/hash/15ddb1773510075ef44981cdb204330b-Abstract-Conference.html
https://dl.acm.org/doi/10.1145/3459637.3482349
https://proceedings.neurips.cc/paper_files/paper/2023/hash/d69103d7895f4e2083f24b664003d386-Abstract-Conference.html

**Theoretical Claims:**

The authors provide a comprehensive theoretical analysis and show consistency of their approach. However, this relies und (potentially strong) assumptions, such as assuming a ground-truth group-structure within the treatments (see more in weaknesses below).

---

> ### Author Rebuttal · Authors · 2025-03-31
>
> We sincerely appreciate your insightful and constructive comments. Below, we have provided our detailed, point-by-point responses.
>
> **1. Behavior under assumption violation**
>
> Our method assumes a ground-truth group structure to establish theoretical guarantees such as consistency of group recovery. This enables identifiability and recovery bounds. While potentially restrictive, it provides a principled foundation for analysis.
>
> In practice, if no exact group structure exists but some treatments have similar effects, the fused lasso encourages grouping of approximately equal effects. If all treatments differ beyond a certain gap, our method will not fuse them. Due to space limit, we refer to
>
> Remark 3.10, which gives the gap tolerance condition; please also see our response to Reviewer UwD5, point 2, for additional context.
>
> **2. Additional experiments and details**
>
> We have added additional simulations for more treatments and under misspecified weight models; see Tables S1 and S2 in the response for Reviewer c8W6 and Table S3 in the response for Reviewer UeTD.
>
> We will move the experimental setup tables to the Appendix.
>
> - Results are averaged over 200 runs; variance is reported in the revised Table S5 below.
> - Policy tree (Zhou et al., 2023) is applied with default tuning.
> - Fused lasso uses extended BIC (line 159) for model selection, a standard criterion for recovering sparse structure.
> - Overall, when no group structure exists, fusion trades variance for bias: it increases effective sample size but may introduce bias if outcome functions differ. Such trade-off is well-studied in multi-source learning and data fusion literature, and can improve overall performance when data are limited.
>
> **Table S5**: Simulation results; CW = Calibration Weighting; ARI (Adjusted Rand Index for fusion quality) and policy value: higher is better; oracle number of groups = 4.
>
> |Method|ARI|Number of Groups|Value|
> |-|-|-|-|
> |policy tree (baseline)|/|16|8.77 (0.08)|
> |fusion + policy tree|0.26 (0.14)|10.725 (1.93)|8.78 (0.09)|
> |CW + fusion + policy tree|**0.96** (0.06)|**4.335** (0.60)|**8.89** (0.11)|
> |Ma et al. (2022)|0.26 (0.14)|10.725 (1.93)|8.51 (0.12)|
> ---
>
> **3. Related work on complex treatments**
>
> We will incorporate the recommended references into the revised manuscript to better position our work within the broader literature.
>
> **4. Presentation**
>
> We revised Section 3.2 to begin with a summary of the key results without technical details, followed by the necessary mathematical intuition and formalism.
>
> **5. Clarification on DGP assumptions**
>
> Our method does not assume a parametric model on the outcome function. Instead, we rely on projections onto a linear working model without assuming the model is correctly specified. This allows our method to remain robust to outcome model misspecification.
>
> On the second bullet point under weaknesses, the sentence after "Second–" appears incomplete. Could you kindly clarify the intended point?
>
> **6. Motivation for calibration weighting and fused lasso over end-to-end approaches**
>
> We adopt a linear fused lasso in the fusion step because of data sparsity prior to grouping. More flexible models like neural networks may overfit in this setting. To mitigate potential model misspecification from using a simple working model, we incorporate calibration weighting, which offers a doubly robust guarantee.
>
> While end-to-end approaches such as neural networks are possible alternatives, they often require larger sample sizes to avoid overfitting and may lack interpretability. In contrast, our method is (i) easy to implement using fused lasso and calibration weighting, (ii) compatible with existing multi-arm policy learning algorithms, and (iii) interpretable, which is especially important in high-stakes applications such as medicine.
>
> **7. Comparison with overlap-based approaches for continuous treatments**
>
> Thank you for highlighting the well-established approach that focuses on treatments with sufficient data (e.g., overlap regions) and the related reference. We will include them in the related work and discussion. This approach is well-suited for continuous, multi-dimensional treatments, such as dosage combinations, and leverages neural networks and constrained optimization to model joint dose-response surfaces. Its strength lies in capturing drug-drug interactions and ensuring reliable off-policy evaluation in such challenging settings.
>
> In contrast, our doubly robust fusion method targets discrete treatments with many levels. In this setting, we are able to utilize information from all regions, not just those with strong overlap. For example, new treatments with few observations but potentially strong effects can be fused and compared to standard-of-care treatments with sufficient historical data. Rather than excluding underrepresented treatments, our method allows for their inclusion through meaningful fusion, enabling a more comprehensive and interpretable evaluation.

---

> > ### Comment · Reviewer_7HRd · 2025-04-06
> >
> > Apologies, I posted the following a few days ago, but as "official comment" (not visible to the authors).
> >
> > Thank you for the rebuttal and the additional experiments. You have addressed my main concerns and I will update my score accordingly. I still think the experimental section is relatively weak, but I am fine with this paper getting accepted given the overall contributions of the paper.

---

> > > ### Author Response · Authors · 2025-04-06
> > >
> > > Thank you very much for your follow-up and for considering our responses and additional experiments. We truly appreciate your constructive feedback.
> > >
> > > To strengthen the experimental section, we will incorporate the additional simulation results with more treatments ($K=32$ and $K=48$) and under misspecified weight models into the revised paper (Tables S1–S3), along with more details about the simulation setup.
> > >
> > > We would be grateful for any further suggestions you might have to improve the experimental section, as your insights would be very helpful for the final version.

---

### Official Review · Reviewer_UwD5 · 2025-03-09

**Overall Recommendation:** 3

**Summary:**

This paper addresses the problem of learning individualized treatment rules (ITR) in the high-dimensional setting when the cardinality of the action space is large. This setting can be challenging given the search space for ITR grows exponentially with the cardinality of the action space. The paper proposes a calibration-weighted fusion approach that robustly groups treatments into a smaller number of groups thereby reducing the effective dimensionality of the action space. The method first uses calibration weighting to re-weight observations so that the covariate distributions in each treatment group resembles the overall population, and then use fused Lasso on a linear model to group treatments whose outcome are similar conditioned on covariates.

The approach is doubly robust: correct group recovery is guaranteed if either the calibration weighting model is correctly specified, or the linear outcome model is correctly specified. After forming these fused groups, the user can then employ standard multi-armed policy-learning methods. The paper shows that the final learned policy is near-optimal with regret bounds. It also provides simulations and a real-world example on CLL and SLL to demonstrate the benefits of the approach in terms of group recovery and policy value optimization.

**Claims And Evidence:**

The paper focuses on the problem of partitioning high dimensional treatments into groups such that treatments within each group have identical conditional outcomes. For this purpose they give a calibration weighting approach that is doubly robust for group recovery, i.e., it can recover the correct grouping if either the calibration or the linear outcome model is correct. Once treatments are grouped, standard multi-armed policy learning methods can be applied to the grouped space with a near-optimal policy regret bound.

The theoretical claims are supported by proofs. But I am a bit skeptical why is it necessary to recover such groups for policy learning policy. For example, there can be scenarios where exact recovery of groups is more costly than directly doing policy learning, or there may be other techniques that does policy learning without full recovery of the groups.

**Essential References Not Discussed:**

n.a.

**Experimental Designs Or Analyses:**

The paper provides empirical results in both synthetic scenarios and real data application.

In the synthetic scenarios, it tests the proposed method on a simulation that involves 16 treatments that form 4 true groups, each group having 4 arms that share the same outcome function and the covariate distribution differs across arms. It shows that using calibration weight, fusion and policy tree obtains the best policy value compared with baselines.

In the real data scenario, it tests on a dataset with 10,346 patients with CLL/SLL with 8 treatment arms and show that the their method with calibration weighting, fusion and policy tree yield 5 groups that implies clinical patterns in the dataset.

**Methods And Evaluation Criteria:**

The paper offers a valuable approach for merging many treatments into fewer groups, thus enabling more effective policy learning techniques for high dimensional settings. The method’s double-robustness is appealing, and the real data application underscores its practicality and interpretability.

**Other Comments Or Suggestions:**

n.a.

**Other Strengths And Weaknesses:**

Strengths:
- The paper extends standard policy learning to handle the high dimensional case with a large number of arms through identifying group structures in among the arms that share the same outcome model. The paper combines calibration weight with fused Lasso for a doubly robust approach.

Weaknesses:
- It seems restrictive that the grouping method only works for groups of action that shares the same outcome function. And it is unclear why one necessarily need to identify all such groups for policy learning purposes.

**Questions For Authors:**

- The paper defines the grouping structure for different actions $a\in A$ when the expected outcome $\mu_a(X)$ are the same. Why does it have to be equality and not approximate equality? And what should be the gap tolerance between different expected outcomes for the algorithm to work?
- Is the grouping necessarily needed for policy learning purposes and is it always the best thing to do to identify this group structure before optimizing? Are there scenarios when, e.g. one of the actions has very high expected outcome, and the other actions have some complicated group structures, but finding this best action doesn't necessarily require identifying the group structure within the suboptimal actions?

**Relation To Broader Scientific Literature:**

The paper extends standard policy learning to handle the high dimensional case with a large number of arms through identifying group structures in among the arms that share the same outcome model. The paper combines calibration weight with fused Lasso for a doubly robust approach.

**Theoretical Claims:**

The paper shows double robustness of the proposed method; the group structure is recovered if either the calibration weighting model is correct or the linear outcome model is correct (Theorem 3.8, 3.11). Minimizing the weighted fused Lasso objective recovers the “oracle” solution that merges arms that share identical $μ_a(x)$. If the grouping is correct, the final policy learning with an M-armed approach achieves an $O(\sqrt{n})$ regret bound (Propositions 3.16, 3.17).

---

> ### Author Rebuttal · Authors · 2025-03-31
>
> We sincerely appreciate your insightful and constructive comments. Below, we have provided our detailed, point-by-point responses.
>
> **1. Necessity of recovering group structure**
>
> When there are many treatments and the sample size is limited, directly performing policy learning presents several challenges: (i) the outcome model cannot be too complex or flexible due to limited data per arm, (ii) covariate shifts across treatment groups exacerbate instability, especially when using balancing methods like IPW, and this instability is amplified for underrepresented arms with small and highly variable propensity scores, and (iii) the theoretical guarantees (e.g., regret bounds) of well-established methods such as multi-arm policy trees (Zhou et al., 2023) typically require the number of treatment levels to be fixed.
>
> By using our doubly robust fusion approach followed by policy learning over the fused treatments, we address these issues: (i) more complex and flexible outcome models can be used due to increased sample sizes within fused groups, (ii) weights can be more stably estimated across fused treatments, and (iii) by reducing the number of treatments, existing multi-arm policy learning methods with theoretical guarantees can be directly applied.
>
> We acknowledge that there are alternative approaches for policy learning without explicit group recovery, such as end-to-end methods mentioned by Reviewer 7HRd. However, we emphasize that our approach (i) avoids potential overfitting issues when sample sizes are small, (ii) is easy to implement using fused lasso and calibration weighting, (iii) is readily compatible with state-of-the-art multi-arm policy learning algorithms, and (iv) offers interpretability, which is especially desirable in high-stakes domains such as medicine.
>
> **2. Approximate equality of $\mu_a(X)$ and gap tolerance**
>
> We define treatment groups based on equality of the expected outcome functions to ensure identifiability and interpretability of the fused structure, which is central to our theoretical analysis. While exact equality may seem restrictive, it serves as a natural starting point for grouping treatments with similar effects and enables formal guarantees.
>
> In practice, our method does tolerate small differences in $\mu_a$ due to sampling variability and regularization. The fused lasso penalization encourages the fusion of approximately equal effects, allowing treatments with similar but not identical outcomes to be grouped together.
>
> We explicitly provide the gap tolerance required for recovery in Remark 3.10. For reference, the condition is:
> $$
> \min\_{b\neq b^\prime} \| \boldsymbol{\beta}\_b^* -\boldsymbol{\beta}\_{b^\prime}^* \|\_{\infty} / c > \lambda\_n \gg \sqrt{n\log(n)}/K,
> $$
> where $\boldsymbol{\beta}\_b^*$ and $\boldsymbol{\beta}\_{b^\prime}^*$ are the group-shared projection vectors from groups $b$ and $b^\prime$, $c$ is a constant, $\lambda_n$ is the penalization level, $n$ is the sample size, and $K$ is the number of treatments.
>
> **3. When is group recovery necessary for policy learning?**
>
> We agree that pre-grouping is not strictly necessary for all policy learning scenarios. In the example the reviewer described, if the primary goal is simply to identify the single best action with the highest expected outcome, then fully recovering the group structure among suboptimal actions may not be essential.
>
> However, in many practical settings, especially in personalized decision-making, the goal is to learn individualized treatment rules rather than selecting a single best action for the entire population. In such cases, identifying meaningful treatment groupings can improve estimation stability, interpretability, and computational tractability, particularly when the action space is large and the sample size is limited. Grouping also allows us to leverage shared structure across similar treatments to reduce variance and enhance overall performance.

---

### Official Review · Reviewer_UeTD · 2025-03-13

**Overall Recommendation:** 4

**Summary:**

The paper proposes to learn an optimal individualized treatment regime in settings with a large number of treatment actions. To handle large action space, the author(s) proposed a novel doubly robust method for grouping treatments into a few categories to enable the application of existing methodologies developed with few treatment options to handle these settings. They further validate their proposed method both theoretically and empirically. Specifically, in theory:

1.  They established the doubly robustness of their group procedure, requiring the correct specification of either the propensity score or the outcome regression model.
2. They proved the oracle property of their estimated $\widehat{\zeta}$.
3. They derived the regret bound of the resulting estimated optimal treatment regime.

Empirically, they conducted simulations and real data analyses to demonstrate the superior finite sample performance of their proposal.

**Claims And Evidence:**

The double robustness, oracle property, and regret bounds are theoretically verified. Numerically, the authors conducted experiments to evaluate the performance of their proposed treatment regime and the (ARI of their grouping method. To further strengthen the claims, it would be beneficial for the authors to conduct additional simulations to empirically validate the double robustness property.

**Essential References Not Discussed:**

I do not think any essential references are missing. However, there is an extensive literature on estimating (dynamic) treatment regimes, and many related papers exist. For further details, please refer to my comments in the "Other Comments & Suggestions" section.

**Experimental Designs Or Analyses:**

As I mentioned earlier, the analysis is theoretically sound and partially supported by numerical experiments. However, one potential limitation of the experimental design is that the current simulation considers only 16 treatment groups. To better address scenarios with truly large action spaces, I would suggest increasing the number of treatment groups to 50 or even 100 in simulations.

**Methods And Evaluation Criteria:**

The paper utilizes a real-world dataset with 7 treatment options, which partially motivates the problem, as the scenario extends beyond the typical two-arm case commonly studied in the literature. However, 7 is not an extraordinarily large number. In fact, the analysis groups these treatments into 5 categories. I would not consider the reduction from 7 to 5 as substantial enough to lead to significant enhancements.

**Other Comments Or Suggestions:**

1. Assumption 2.1(i) Should be the consistency assumption, instead of SUTVA. SUTVA requires the potential outcome depends on its own treatment value, which differs from the consistency which requires the observed outcome to equal the potential outcome.

2. I will not include $\mathbb{P}(X=x)>0$ in Assumption 2.1(iii). It is not the typical positivity assumption referred to in the literature.

3. Page 2, K is probably not the treatment dimension, but the treatment cardinality. Similarly, the title refers to "high-dimensional treatments," but it seems more appropriate to use "high-cardinality treatments."

4. The first part of Assumption 3.1 is more like a claim than an assumption. Would it be possible to provide a proof for this rather than assuming it holds?

5. You imposed the bounded outcome and covariates only in Section 3.3. Does this imply that the theories in Section 3.2 do not rely on this assumption?

6. Propositions 3.16 and 3.17 seem quite general. Do they truly depend on specifying the policy class as trees? If not, I suggest revising the title of this section and placing less emphasis on policy trees to better reflect the generality of the results.

7. As I mention earlier, there is a large literature on individualized (dynamic) treatment regimes. The author(s) might want to include more relevant papers. For instance:

    * Q-learning introduced in the introduction is adapted from the well-known Q-learning algorithm by Watkins https://link.springer.com/article/10.1007/BF00992698. There are also follow-up works on dynamic treatment regimes; see e.g., https://www3.stat.sinica.edu.tw/sstest/j25n3/j25n35/j25n35.html.
    * In addition to the work by Zhang et al. (2012), outcome-weighted learning https://pmc.ncbi.nlm.nih.gov/articles/PMC3636816/ also belongs to the value-search method. Linear ITRS have been explored thoroughly in the literature, including in Zhang et al. (2012), the outcome-weighted learning paper, and e.g., https://pmc.ncbi.nlm.nih.gov/articles/PMC5966293/.
    * Similarly, tree-type methods have been explored very early by https://pubmed.ncbi.nlm.nih.gov/26893526/ and  https://arxiv.org/abs/1504.07715. It is unclear why the authors cite Athey and Wager’s paper as the sole representative of this line of research.
    * Finally, https://proceedings.neurips.cc/paper_files/paper/2021/file/816b112c6105b3ebd537828a39af4818-Paper.pdf also studies the grouping of actions, but in a continuous action space.

**Other Strengths And Weaknesses:**

I have discussed the strengths of the paper in the "Summary" and "Relation To Broader Scientific Literature" Sections. Some potential weakness are given below:

1. As I mentioned, one strength of the paper lies in that they do not rely on the linearity assumption. However, this comes at a price for their grouping method. Specifically, their identified groups might be different from the oracle groups. This is because when the oracle groups are the same, it ensures that their identified groups are asymptotically the same, based on the equation above (9). However, the converse is not true. I was wondering if the author(s) can impose some completeness-type assumptions to guarantee the reverse also holds.

2. Another potential limitation lies in the presentation. The paper relies heavily on notations and mathematical expressions, which could be streamlined. It would be ideal if the authors could reduce the number of notations and replace unnecessary formulas with words. For example, the discussion following Theorem 3.8 is intended to explain the theorem, but it itself involves several mathematical notations, making it less accessible. Similarly, Assumption 3.9 is highly technical, and Remark 3.10, which aims to clarify it, remains quite technical as well. Some of the assumptions and notations could be simplified. For instance, $w_i^*$ essentially represents the inverse propensity score, so the second part of Assumption 3.1 is automatically satisfied due to the positivity assumption and does not need to be explicitly imposed. Likewise, Assumption 3.2 fundamentally requires the correct specification of the propensity score, which could be stated more straightforwardly.

**Questions For Authors:**

Please refer to the section above.

**Relation To Broader Scientific Literature:**

The paper makes valuable contributions that extend beyond the existing literature. Scientifically, large action spaces have been relatively underexplored in the context of individualized (dynamic) treatment regimes. Methodologically, the proposed approach is novel and supported by theoretical guarantees, including the proven double robustness property. A particularly appealing feature of the method is that it does not rely on linear model assumptions. Instead, it uses linear models solely as working models to facilitate the application of fused lasso for grouping treatments. This represents a significant advancement over prior work, such as Ma et al. (2022).

**Theoretical Claims:**

The theories appear reasonable to me, though I did not check the proofs in detail.

---

> ### Author Rebuttal · Authors · 2025-03-31
>
> We sincerely appreciate your insightful and constructive comments. Below, we have provided our detailed, point-by-point responses.
>
> **1. Simulations for validation of the double robustness**
>
> We additionally considered a scenario where the outcome mean functions are linear (correctly specified), while the weighting model is misspecified. Specifically, for calibration weighting, we only used $X_2$ and $X_3$, omitting $X_1$.
>
> Since the outcome models are correctly specified, both the *fusion* and *CW + fusion* approaches achieved strong ARI scores, demonstrating the double robustness property. Ma et al. (2022) also performed well since the linearity assumption they rely on holds in this setting. However, our proposed method still achieved the best performance.
>
> Note that the simulation in Section 4.1 already showed the advantage of the proposed method when the outcome model is misspecified while the weighting model is correct.
>
> **Table S3**: Simulation results under misspecified weighting model; CW = Calibration Weighting; ARI (Adjusted Rand Index for fusion quality) and policy value: higher is better; oracle number of groups = 4.
>
> |Method|ARI|Number of Groups|Value|
> |-|-|-|-|
> |policy tree (baseline)|/|16|6.35 (0.06)|
> |fusion + policy tree|0.88 (0.13)|5.42 (1.42)|6.41 (0.04)|
> |CW + fusion + policy tree|**0.96** (0.06)|**4.46** (0.66)|**6.43** (0.02)|
> |Ma et al. (2022)|0.88 (0.13)|5.42 (1.42)|6.39 (0.00)|
> ---
>
> **2. Real-world experiments**
>
> The real-world dataset motivates our treatment fusion approach due to practical challenges such as limited sample size per treatment arm. We acknowledge that the grouping from 7 to 5 is determined by the underlying structure of the real data.
>
> To further illustrate the effectiveness of our method in larger treatment spaces, our simulation study demonstrates its ability to reduce 16 treatment arms to 4 groups. We also consider scenarios with more treatments in the next response point.
>
> **3. Simulation with more treatments**
>
> We increased the number of treatments to 32 and 48 while keeping the total sample size fixed. We refer to Tables S1 and S2 in the response for Reviewer c8W6, point 1, for results. The proposed method continues to outperform other methods.
>
> **4. Completeness assumption**
>
> To guarantee recovery of the oracle grouping, we additionally imposed the completeness assumption: for any function $h(\cdot)$, if $\mathbb{E}\\{X h(X)\\}=0$, then $h(X)=0$ almost surely.
>
> **5. Presentation**
>
> We will streamline the manuscript by adding plain-language explanations (e.g., after Theorem 3.8) and moving heavy notation and technical content to the appendix to enhance accessibility.
>
> While $w^*$ equals the inverse propensity score when the model is correct, it is not in general, as we allow model misspecification; thus, the second part of Assumption 3.1 is retained.
>
> We will revise Assumption 3.2 based on your suggestion.
>
> **6. Assumption 2.1 (Identification)**
>
> We replaced “SUTVA” with the consistency and removed $P(X = x) > 0$.
>
> **7. Title terminology**
>
> We revised the title and text to use “many treatments” instead of “high-dimensional treatments,” following conventions in the literature (Ma et al., 2022, 2023).
>
> **8. Assumption 3.1 (Convergence of calibration weight)**
>
> For every treatment $a\in\mathcal{A}$, calibration weighting is an optimization problem and can be solved using the method of Lagrange multipliers:
> $$
> L\_a(w\_1,\ldots,w\_n)=\sum_{i:A\_i=a}\\{\gamma(\gamma+1)\\}^{-1}\\{(n\_a w\_i)^{\gamma+1}-1\\}-n\lambda^\top \sum\_{i:A\_i=a}w\_i(X_i-\bar{X})+n\varphi\left(1-\sum_{i:A\_i=a} w_i\right).
> $$
> Minimizing $L\_a(w\_1,\ldots,w\_n)$ gives:
> $$
> \hat{w}\_i=w(X_i;\hat\lambda)=\frac{\rho\_{\gamma}[\hat\lambda^\top(X_i-\bar{X})]}{\sum\_{j:A_j=a} \rho\_{\gamma}[\hat\lambda^\top(X_j-\bar{X})]},
> $$
> where the function $\rho\_{\gamma}(x)$ for different $\gamma$ values are summarized in Table S4 below, and $\widehat{\lambda}$ solves the equation
> $$
> \sum\_{i:A\_i=a}\rho\_{\gamma}\[\lambda^\top(X\_i-\bar{X})\](X\_i-\bar{X})=0.
> $$
> Therefore, $\hat{\lambda}$ is an M-estimator and, under standard regularity conditions for M-estimators (Boos and Stefanski, 2013), it is root-n consistent.
>
> **Table S4**: $\rho\_{\gamma}(x)$ for Cressie-Read family.
>
> |$\gamma$|$h\_{\gamma}(w)$|$\rho\_{\gamma}(x)$|
> |-|-|-|
> |$-1$|$-\ln(nw)$|$(1 - x)^{-1}$|
> |$0$|$nw\ln(nw)$|$\exp(x)$|
> |$\gamma$|$\frac{(nw)^{\gamma+1} - 1}{\gamma(\gamma+1)}$|$(1 + \gamma x)^{1/\gamma}$|
> ---
>
> **9. Bounded assumption**
>
> The bounded outcome and covariates assumption is only required for the results in Section 3.3 and not for those in Section 3.2.
>
> **10. Generality of results**
>
> Propositions 3.16 and 3.17 do not rely on the policy class being restricted to trees. We revised the section title to “Multi-armed Policy Learning” to reflect the broader applicability.
>
> **11. Related literature**
>
> We will incorporate the recommended references into the revised manuscript to better situate our work within the broader literature.

---

### Official Review · Reviewer_c8W6 · 2025-03-25

**Overall Recommendation:** 3

**Summary:**

The authors study causal inference with many treatments and sparse data within each group, which makes estimating treatment effects challenging. They observe that many treatments share commonalities and can be clustered to reduce the effective dimensionality of the treatment space. Within each group, treatments are treated as equivalent, allowing the application of efficient multi-armed ITR learning methods. The authors propose calibration-weighted treatment fusion, a technique that leverages balancing weights in a regression model to estimate the group structure of treatments in a doubly robust manner. Using these fused groups, they propose AIPW estimators for policy learning.

**Claims And Evidence:**

Theoretical claims are supported by theorems and proofs. The authors provide formal guarantees for consistent estimation of the counterfactual mean parameters of treatments and recovery of fused treatment groups. They then provide regret bounds for a Cross-Fitted AIPW Policy Learning algorithm that leverages their fused treatment approach.

**Essential References Not Discussed:**

Some relevant work worth mentioning is [2] and the references therein on synthetic combinations, which provide a synthetic control-like framework for estimating counterfactual means using low-rank matrix completion when the number of possible treatments is large or combinatorially explosive.



[2] Agarwal, Abhineet, Anish Agarwal, and Suhas Vijaykumar. "Synthetic combinations: A causal inference framework for combinatorial interventions." Advances in Neural Information Processing Systems 36 (2023): 19195-19216.

**Experimental Designs Or Analyses:**

The experimental design looked well thought out. However, as mentioned in Methods And Evaluation Criteria, it would be useful to see how the method performs in a more diverse range of scenarios characterizing by data sparsity and number of treatments.

**Methods And Evaluation Criteria:**

The authors evaluate their approach in synthetic and real-world datasets against reasonable benchmarks. Their method performs well in the simulations and the real-world dataset highlights the practical applicability and interpretability of the method in policy learning.

It would be beneficial to provide experiments investigating the performance of the method as the number of treatments increases for a fixed sample size, with the number of observations per treatment thus decreasing. Does the method become unstable? What regimes would the method be expected to work well in, and where might it not perform well?

**Other Comments Or Suggestions:**

See other parts for comments.

**Other Strengths And Weaknesses:**

Strength: The approach tackles a notable problem in causal inference with many treatments, where treatment groups may be sparse and have insufficient overlap. The idea of grouping similar treatments to reduce the dimensionality of the treatment space provides a practical solution. A nice feature of the proposed method is that once these groups are learned using their method, standard causal inference and policy learning methods can then be used with these fused treatments.

Weaknesses: The method requires estimation of balancing weights within each treatment arm. Since each treatment arm may contain very few observations, it seems that this could be quite challenging to estimate well. For example, in some settings, as considered in [2], some treatment arms may have very few or even zero observations. In such cases, I imagine the authors' technique may not apply or may perform poorly.

**Questions For Authors:**

When learning the balancing weights, would it be beneficial to use weights for the fused treatments, such that more data can be used to learn them? For instance, one could use an iterative approach that first learns a preliminary fused treatment and then estimates the calibrated weights for the fused treatments. This may allow the application of this method when treatment arms have very few observations such that learning weights is infeasible.

**Relation To Broader Scientific Literature:**

Related literature is discussed.

**Theoretical Claims:**

The theoretical claims seem correct but I need not check the proofs.

Assumption 3.1 requires parametric-rate estimation of the weights. Is this necessary? Since the estimator is doubly robust with respect to the outcome and weight models, I would expect that a weaker condition, such as faster than n^{-1/4} convergence of nuisances, might suffice, e.g., as in [1].


[1] Chambaz, Antoine, Pierre Neuvial, and Mark J. van der Laan. "Estimation of a non-parametric variable importance measure of a continuous exposure." Electronic journal of statistics 6 (2012): 1059.

---

> ### Author Rebuttal · Authors · 2025-03-31
>
> We sincerely appreciate your insightful and constructive comments. Below, we have provided our detailed, point-by-point responses.
>
> **1. Experiments for increasing $K$ and fixed $n$**
>
> We keep the sample size fixed at $n=1800$ and increase the number of treatments $K$ from 16 to 32 and 48. In such regime, our proposed method outperforms other baselines in terms of fusion quality and policy value. The number of recovered groups increases slightly, as expected. Our method may become unstable when the number of treatments approaches the sample size, leaving very few observations per arm. In such regimes, iterative weight learning, as suggested by the reviewer, may help stabilize performance (see point 5).
>
> ---
> **Table S1**: Simulation results for $K=32$; CW = Calibration Weighting; ARI (Adjusted Rand Index for fusion quality) and policy value: higher is better; oracle number of groups = 4.
>
> |Method|ARI|Number of Groups|Value|
> |-|-|-|-|
> |policy tree (baseline)|/|32|8.66 (0.18)|
> |fusion + policy tree|0.21 (0.09)|16.74 (3.64)|8.63 (0.18)|
> |CW + fusion + policy tree|**0.85** (0.10)|**5.68** (1.72)|**8.80** (0.21)|
> |Ma et al. (2022)|0.21 (0.09)|16.74 (3.64)|8.52 (0.12)|
> ---
> **Table S2**: Simulation results for $K=48$.
>
> |Method|ARI|Number of Groups|Value|
> |-|-|-|-|
> |policy tree (baseline)|/|48|8.49 (0.20)|
> |fusion + policy tree|0.16 (0.08)|21.00 (5.58)|8.40 (0.30)|
> |CW + fusion + policy tree|**0.74** (0.10)|**7.35** (2.31)|**8.52** (0.23)|
> |Ma et al. (2022)|0.16 (0.08)|21.00 (5.58)|8.41 (0.12)|
> ---
>
> **2. Convergence rate of the weights**
>
> In the fusion phase, Assumption 3.1 posits parametric-rate estimation of the weights to ensure consistency of the estimated grouping under outcome model misspecification. Specifically, if the weight model is correctly specified and estimated at the parametric rate, our method can tolerate full misspecification of the linear working outcome model, which is an appealing feature in settings with many treatment arms and limited per-arm data. In contrast, allowing slower convergence for the weight estimator (e.g., $o(n^{-1/4})$) would require the development of doubly robust estimators and the estimation of an additional nuisance model, specifically the true conditional mean function of the outcome, at a rate of at least $O(n^{-1/4})$. Such doubly robust estimators are appealing when there is a large sample size per treatment arm. However, in our setting with limited data per arm, more flexible models may result in overfitting and unstable performance.
>
> In the policy learning phase, Assumption 3.1 is no longer required for the weight model. Instead, we adopt Assumption 3.13, which imposes a weaker rate double robustness condition, as the reviewer suggested. This relaxation is justified by the increased sample size within each fused group, enabling the use of more flexible models for both the weight and outcome components.
>
> **3. Synthetic combinations**
>
> We clarify that our setting involves $K$ treatment levels without combinatorial structure, while [2] considers $2^p$ combinations from $p$ binary treatments. To reflect this distinction, we will revise our title to “Doubly Robust Fusion of Many Treatments for Policy Learning,” following Reviewer UeTD’s suggestion and literature conventions (Ma et al., 2022, 2023).
>
> While our method uses calibration-weighted fused lasso to reduce the action space for policy learning, synthetic combinations aim to impute counterfactuals under structural assumptions. These are complementary strategies, and we added a discussion in the revised manuscript.
>
> **4. Arms with few or even zero units**
>
> If some treatment arms have very few or even zero observations, there is little information for those arms, and our method might not be stable in such cases unless additional structural assumptions across the $K$ treatments are imposed, such as the combinatorial structure used in [2]. The instability of learned balancing weights in these challenging scenarios highlights the difficulty of the problem and motivates potential methodological refinements. For example, one possible solution is to conduct iterative weighting and fusion, as discussed in the next point.
>
> **5. Iterative learning of weights for fused treatments**
>
> It is beneficial to estimate weights for the fused treatments, as there might be more data in each treatment group than before fusion, which is also our main motivation for performing fusion. See also point 2 regarding the convergence rate of the weights for the fusion and policy learning phases.
>
> In the fusion phase, while our current approach performs a single round of fusion, we appreciate the reviewer’s suggestion of iterative learning of weights to refine the preliminary fusion and discussed this extension to handle challenging cases with few observations in some treatment arms in the manuscript.
>
> In the policy learning phase, we indeed re-estimate the weights for the fused treatments when learning the individualized treatment rule.

---

> > ### Comment · Reviewer_c8W6 · 2025-04-04
> >
> > Thank you for your comments and I appreciate the additional simulations. I maintain my score.
> >
> > ### 2.
> > I view the omission of doubly robust methods as a limitation of the work. At a minimum, the approach should be discussed, and any limitations should be clearly articulated in the main text.
> >
> > Doubly robust methods can also be used with simple models. In general, doubly robust estimators reduce bias relative to IPW and plug-in methods. When the weights are estimated at rate $O_p(n^{-1/2})$ (as currently assumed), the outcome regression only needs to be consistently estimated—at an arbitrarily slow rate—for asymptotic normality to hold. The outcome regression can be estimated by the empirical mean of the outcome within treatment arms. Even with small sample sizes, this should still provide some bias reduction.
> >
> >
> > Furthermore, when weights are estimated nonparametrically or with undersmoothing, existing results show that IPW-based or balancing weight estimators are debiased and asymptotically equivalent to doubly robust estimators [1,2]. Such results may be relevant in the current setting, particularly when calibrated weights are estimated without regularization.
> >
> >
> >
> > [1] Bruns-Smith, D., Dukes, O., Feller, A., & Ogburn, E. L. (2023). Augmented balancing weights as linear regression. arXiv preprint arXiv:2304.14545.
> > [2] Ertefaie, Ashkan, Nima S. Hejazi, and Mark J. van der Laan. "Nonparametric inverse‐probability‐weighted estimators based on the highly adaptive lasso." Biometrics 79.2 (2023): 1029-1041.

---

> > > ### Author Response · Authors · 2025-04-05
> > >
> > > We sincerely thank the reviewer for this valuable comment on point 2 and detailed explanation with references. We fully agree that doubly robust methods, as the reviewer illustrated, can be applied with simple models and help relax the convergence rate requirement for the weights. We will clearly articulate this point in the main text and include a discussion of more advanced doubly robust methods and double machine learning approaches in the fusion stage, which can further relax convergence rate requirements for both the weight and outcome models.

---

### Decision · Program_Chairs · 2025-05-01

**Decision:**

Accept (poster)

**Comment:**

I concur with the reviewers' positive view of the submission and think the paper would make a nice addition. Several points for revision and improvement were brought up in the discussion (thanks to all for participating actively) and I fully expect the authors to incorporate these into the final paper. A couple points were left unresponded that can also be addressed (such as discussion of completeness and oracle grouping).